# Aβ efflux impairment and inflammation linked to cerebrovascular accumulation of amyloid-forming amylin secreted from pancreas

Nirmal Verma[1,2,23], Gopal Viswanathan Velmurugan[1,23], Edric Winford[1,3], Han Coburn[1], Deepak Kotiya[1,2], Noah Leibold[1,2], Laura Radulescu[1,2], Sanda Despa[1,2], Kuey C. Chen[1,4], Linda J. Van Eldik [5], Peter T. Nelson[5], Donna M. Wilcock[5,6], Gregory A. Jicha[5,7], Ann M. Stowe[7], Larry B. Goldstein [7], David K. Powel[8], Jeffrey H. Walton [9], Manuel F. Navedo [10], Matthew A. Nystoriak [11], Andrew J. Murray [12], Geert Jan Biessels [13], Claire Troakes [14], Henrik Zetterberg[15,16,17,18], John Hardy[17,18,19,20,21], Tammaryn Lashley[17,22] & Florin Despa [1,2,3,7 ✉]

Impairment of vascular pathways of cerebral β-amyloid (Aβ) elimination contributes to Alzheimer disease (AD). Vascular damage is commonly associated with diabetes. Here we show in human tissues and AD-model rats that bloodborne islet amyloid polypeptide (amylin) secreted from the pancreas perturbs cerebral Aβ clearance. Blood amylin concentrations are higher in AD than in cognitively unaffected persons. Amyloid-forming amylin accumulates in circulating monocytes and co-deposits with Aβ within the brain microvasculature, possibly involving inflammation. In rats, pancreatic expression of amyloid-forming human amylin indeed induces cerebrovascular inflammation and amylin-Aβ co-deposits. LRP1-mediated Aβ transport across the blood-brain barrier and Aβ clearance through interstitial fluid drainage along vascular walls are impaired, as indicated by Aβ deposition in perivascular spaces. At the molecular level, cerebrovascular amylin deposits alter immune and hypoxia-related brain gene expression. These converging data from humans and laboratory animals suggest that altering bloodborne amylin could potentially reduce cerebrovascular amylin deposits and Aβ pathology.

[1] Department of Pharmacology and Nutritional Sciences, University of Kentucky, Lexington, KY, USA. [2] The Research Center for Healthy Metabolism, University of Kentucky, Lexington, KY, USA. [3] Department of Neuroscience, University of Kentucky, Lexington, KY, USA. [4] UKHC Genomics Laboratory, University of Kentucky, Lexington, KY, USA. [5] Sanders-Brown Center on Aging, University of Kentucky, Lexington, KY, USA. [6] Department of Physiology, University of Kentucky, Lexington, KY, USA. [7] Department of Neurology, University of Kentucky, Lexington, KY, USA. [8] Magnetic Resonance Imaging and Spectroscopy Center, University of Kentucky, Lexington, KY, USA. [9] NMR Facility, University of California, Davis, CA, USA. [10] Department of Pharmacology, University of California, Davis, CA, USA. [11] Department of Medicine, University of Louisville, Louisville, KY, USA. [12] Department of Physiology, Development and Neuroscience, University of Cambridge, Cambridge CB2 3EG, UK. [13] Department of Neurology, University Medical Center Utrecht, Utrecht, The Netherlands. [14] Basic and Clinical Neuroscience Department, King's College London, London, UK. [15] Department of Psychiatry and Neurochemistry, Institute of Neuroscience and Physiology, The Sahlgrenska Academy at the University of Gothenburg, Mölndal, Sweden. [16] Clinical Neurochemistry Laboratory, Sahlgrenska University Hospital, Mölndal, Sweden. [17] Department of Neurodegenerative Disease, UCL Queen Square Institute of Neurology, Queen Square, London WC1N 3BG, UK. [18] UK Dementia Research Institute at UCL and Department of Neurodegenerative Disease, UCL Institute of Neurology, University College London, London, UK. [19] Reta Lila Weston Institute, UCL Queen Square Institute of Neurology, 1 Wakefield Street, London WC1N 1PJ, UK. [20] UCL Movement Disorders Centre, University College London, London, UK. [21] Institute for Advanced Study, The Hong Kong University of Science and Technology, Hong Kong SAR, China. [22] Queen Square Brain Bank for Neurological Disorders, Department of Clinical and Movement Neuroscience, UCL Queen Square Institute of Neurology, London, UK. [23] These authors contributed equally: Nirmal Verma, Gopal Viswanathan Velmurugan. ✉email: f.despa@uky.edu

Alzheimer disease (AD) is characterized by overexpression and/or impaired clearance of Aβ that can be related to an early-onset genetic predisposition to Aβ pathology (familial AD) or occur sporadically with age (sporadic AD)[1]. The factors that balance effective elimination versus accumulation of Aβ have not been fully defined. Well-established pathways of brain Aβ clearance include the interstitial fluid drainage along the walls of cerebral blood vessels, transport across the blood-brain barrier (BBB), and metabolism by microglia and perivascular macrophages[2–4].

Amylin (also known as islet amyloid polypeptide) is a pancreatic β-cell hormone co-released with insulin[5], crosses the BBB[6], and is involved in the central regulation of satiation[7]. In persons with type-2 diabetes mellitus (type-2 DM), amylin forms pancreatic amyloid[8–10] (>95% prevalence at autopsy)[10] and is associated with pancreatic inflammation[11–13]. Data from different research teams show that amylin synergistically co-aggregates with Aβ within brain parenchymal tissue and also deposits within brain arterioles and capillaries, in the settings of both sporadic and familial AD[14–24]. Higher concentrations of pancreas-derived amylin in the central nervous system are associated with an increased frequency of cognitive impairment[21–24]. In APPswe/PS1dE9 (APP/PS1) rats, pancreatic expression of human amylin (rodent amylin is non-amyloidogenic) accelerates behavior deficits and brain Aβ deposition[21]. In rats without Aβ pathology (the HIP rat), pancreatic expression of amyloid-forming human amylin promotes cerebrovascular amylin deposits leading to neuroinflammation[18,25,26] and neurological deficits[18,21,25]. These data form the basis of our hypothesis that increased concentrations of amyloid-forming amylin in the blood promote cerebrovascular amylin deposition and are critical contributing factors to perivascular inflammation and disrupted Aβ clearance in AD. To determine a potential association between pancreas-derived amylin in blood and impaired brain Aβ clearance, we measured amylin concentrations in blood from humans with AD-type dementia versus cognitively unimpaired individuals and assessed the relationships with brain parenchymal and vascular Aβ. To understand the mechanism and uncover novel therapeutic targets for reducing/preventing the development of brain Aβ pathology, we performed comparative pathophysiological characterizations of brain Aβ clearance pathways in transgenic rats expressing amyloid-forming human amylin in the pancreas versus control rats that express endogenous, non-amyloidogenic rat amylin. The results of this study could help better understanding of how altering bloodborne amylin may be used as a therapeutic strategy to potentially reduce cerebrovascular amylin deposits and Aβ pathology.

## Results

**Pancreatic amyloid-forming amylin accumulates in blood and circulating monocytes.** The overall hypothesis tested in our study along with work flow and methods are graphically described in Fig. 1a, b. Using ELISA, we measured amylin concentrations in blood samples collected from cognitively unimpaired individuals (CU; $n = 42$) and persons with sAD-type dementia (DEM; $n = 19$) or mild cognitive impairment (MCI; $n = 19$) (see Table 1 for summary statistics for age and sex). Groups had similar blood glucose concentrations ($112.9 \pm 5.71$ mg/dL vs. $119.1 \pm 9.43$ mg/dL vs. $113.2 \pm 5.10$ mg/dL; one-way ANOVA, $P = 0.79$) and age ($79.35 \pm 2.18$ years vs. $81.35 \pm 1.78$ years vs. $77.60 \pm 0.66$ years; one-way ANOVA, $P = 0.14$). Descriptive statistics of blood amylin concentrations are shown in Supplemental Fig. S1a. Blood amylin concentrations were higher in DEM vs. CU groups (Fig. 1c) (Kruskal-Wallis one-way analysis of variance, $P < 0.001$). In groups divided based on type-2 DM status, blood amylin

concentrations were highly variable (Supplemental Fig. S1b), which may reflect effects of anti-diabetic drugs. A potential link between increased blood amylin concentrations and diabetes was further assessed by measuring the amylin-insulin relationship in the same blood samples as in Fig. 1c. Insulin and amylin ELISAs show that increased blood insulin concentrations are associated with greater blood amylin concentrations ($r = 0.52$; $P < 0.0001$) (Fig. 1d and Supplemental Fig. S1c). The pairwise correlation coefficient demonstrates that hyperamylinemia and hyperinsulinemia are correlated in AD.

Because increased secretion of amyloid-forming amylin promotes amylin accumulation in macrophages and dendritic cells within pancreatic islets[11–13], we hypothesized that chronically elevated blood amylin levels trigger systemic inflammation. Using flow cytometry, we sorted fractions of circulating CD14+ monocytes positive for amylin. We quantified fractions of circulating monocytes positive for amylin ($Q_2$) and those negative for amylin ($Q_1$) in blood samples with amylin concentrations in the upper quartile (>3.5 pM) and in those with amylin concentrations in the lower quartile (<1.5 pM). Blood samples with amylin concentrations in the upper quartile (>3.5 pM) contained increased fractions of CD14+ monocytes positive for amylin ($Q_2$) (Fig. 1e). Confocal microscopic imaging confirmed amylin inclusions in circulating CD14+ monocytes (Fig. 1f).

**Amylin and Aβ co-aggregate within the brain microvasculature, in persons with AD.** We next aimed to delineate possible associations between increased blood amylin concentration and amylin and Aβ accumulation in the brain. Using ELISA, we measured concentrations of amylin and Aβ42 in temporal cortex homogenates from persons with sAD identified by well-established neuropathological characteristics[27–31] ($n = 42$; $n = 22$ with type-2 DM) and from individuals without sAD pathology ($n = 18$; $n = 6$ with type-2 DM) (see Table 2 for clinical data). Individuals are age-similar ($85.65 \pm 7.04$ years vs. $87.22 \pm 7.30$ years), in the sAD vs. non-AD groups. Brain amylin concentrations were higher in persons from the sAD compared to those in the control group (unpaired $t$ test, $P < 0.01$) (Supplemental Fig. S1d), consistent with previous results from other cohorts[15,21]. There was no difference in brain amylin levels between those with type-2 DM versus those without (Supplemental Fig. S1e); however, the analysis did not control for the potential effects of glycemic drugs nor for medications given to individuals with cognitive impairment. Increased brain amylin concentrations were associated with greater Aβ42 concentrations ($r = 0.34$; $P < 0.05$) (Fig. 1g), consistent with the amylin-Aβ42 relationship recently identified in fAD brains[21]. Using matched plasma and brain tissue homogenates that were available in this cohort ($n = 12$ in the sAD group $n = 8$ controls), we assessed the relationship between antemortem plasma amylin concentrations and amylin concentrations in autopsied brain tissue. The pairwise correlation coefficient suggests a possible relationship between circulating amylin levels and the propensity of amylin to accumulate in the brain ($r = 0.40$; $P = 0.09$) (Fig. 1h) (analysis excluded potential outliers of brain tissue amylin concentrations; Supplemental Fig. S1e). The small sample size of matched plasma and brain samples is a limitation.

To test potential overlapping of amylin and Aβ at the BBB, we analyzed sAD and fAD brains for histological evidence of vascular amylin-Aβ co-localization by using immunohistochemistry (IHC), confocal microscopy and proximity ligation assay (PLA) with anti-amylin and anti-Aβ antibodies. For deconvolution and analysis of amylin immunoreactivity intensity signals on IHC images, pancreatic tissue from a patient with type-2 DM served as a positive control for amylin deposition (Supplemental Fig. S1f).

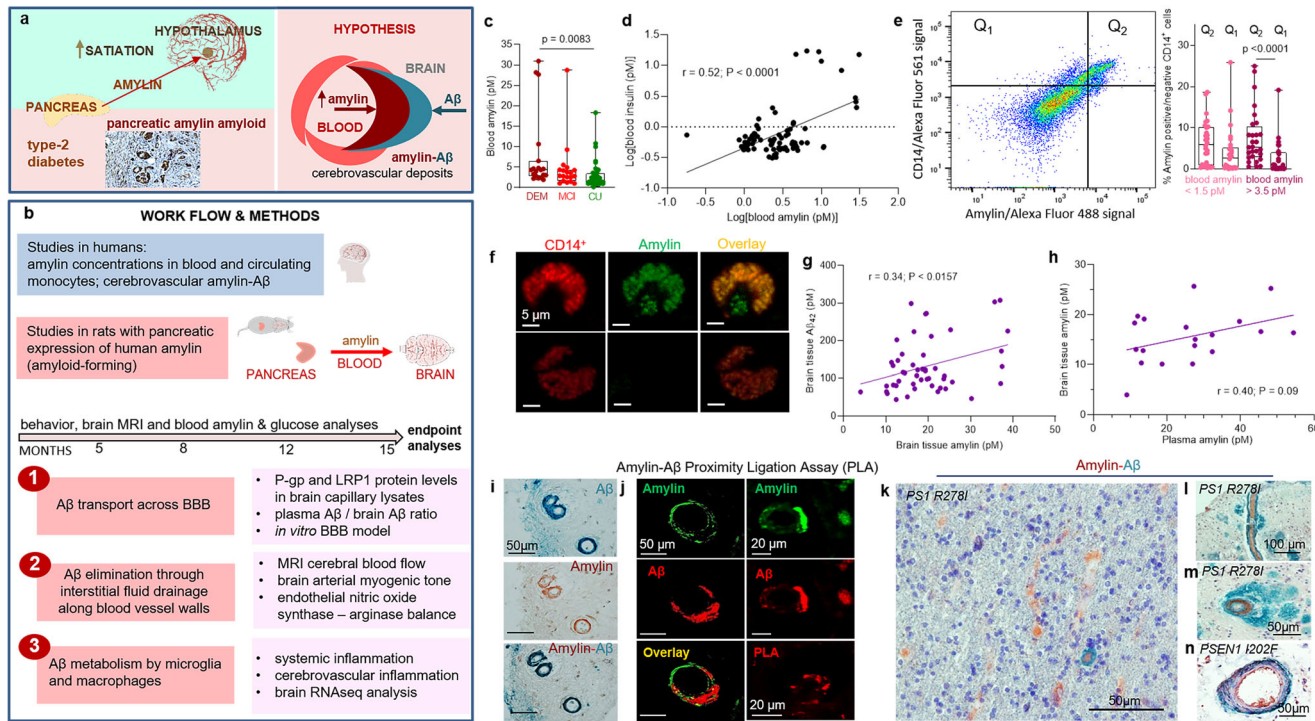

**Fig. 1 The hypothesis of amylin-induced impairment of brain Aβ clearance.** Putative amylin function (green panel) and pathology (red panels) (**a**) along with work flow and methods (**b**). **c** Blood amylin concentrations in dementia (DEM; $n = 19$), mild cognitive impairment (MCI; $n = 19$), and cognitively unimpaired (CU; $n = 42$) individuals. **d** Pairwise correlation coefficient ($r$) between amylin and insulin concentrations in same blood samples as in (**c**). **e** Flow cytometry sorting of amylin positive ($Q_2$) or negative ($Q_1$) circulating $CD14^+$ monocytes in blood with lower quartile vs. upper quartile amylin concentrations. **f** Confocal microscopic images showing amylin engulfed in $CD14^+$ monocytes ($n = 3$). Pairwise correlation coefficient ($r$) between amylin and $Aβ_{42}$ concentrations in human brains, including persons with sAD ($n = 42$) and without AD ($n = 18$) (**g**), and between matched antemortem plasma amylin concentration and amylin concentration in autopsied brain tissue, including persons with sAD ($n = 12$) and without AD ($n = 8$) individuals (**h**) (potential outliers were removed from the analysis; shown in Supplemental Fig. S1e). **i** IHC analysis using anti-amylin (brown) and anti-Aβ (green) antibodies on serial sections from a sAD brain ($n = 18$). **j** Confocal microscopic analysis and amylin-Aβ proximity ligation assay (PLA) showing vascular amylin-Aβ deposits in a sAD brain. IHC analysis of fAD brains showing Aβ deposits in perivascular spaces and vessel walls with amylin accumulation within the lumen (**k**, **l**), and amylin deposits in the vessel wall (**m**) or vessel wall (**n**), and Aβ deposits in perivascular spaces ($n = 32$). Data are presented as box and whiskers or correlation analyses; Kruskal–Wallis one-way of variance, Data are means ± SEM.

**Table 1 Summary statistics for age, sex and type-2 DM status in individuals included in the cross-sectional analyses of: 1, amylin in blood and circulating monocytes; and 2, brain amylin and $Aβ_{42}$ concentrations by using ELISA.**

|  | DEM | MCI | CU |
|---|---|---|---|
| **Blood samples** | $n = 19$ | $n = 19$ | $n = 42$ |
| Sex, female/male (% female) | 9/10 (47.4%) | 14/4 (77.8%) | 20/21 (48.8%) |
| Age at collection (avg ± SEM) | 79.35 ± 2.18 | 81.35 ± 1.78 | 77.60 ± 0.66 |
| Type-2 diabetes | 6 (28.5%) | 5 (25%) | 6 (14.28%) |
|  | **DEM** | **CU** |  |
| **Frozen brain tissue** | $n = 42$ | $n = 18$ |  |
| Sex, female/male (% female) | 25/17 (59.5%) | 14/4 (77.8%) |  |
| Age at collection (avg ± SEM) | 85.65 ± 7.04 | 87.22 ± 7.3 |  |
| Type-2 diabetes | 22 (52.4%) | 6 (33.3%) |  |

Histopathological analysis of cerebrovascular amylin-Aβ co-localization is summarized in Table 3. Representative images from serial sectioning and IHC staining with anti-amylin, anti-Aβ and combined anti-amylin and anti-Aβ antibodies in temporal cortex tissues from an 83-years old woman with sAD and type-2 DM are shown in Fig. 1i. Confocal microscopic analysis and PLA with the same anti-amylin and anti-Aβ antibodies further confirmed cerebrovascular amylin-Aβ deposition (Fig. 1j). The PLA signal shows an overall consistency with amylin-Aβ colocalization appearing within the arteriolar wall. For comparison, images of IHC analysis of temporal cortex tissue from an 86-years old cognitively unimpaired woman without type-2 diabetes are shown in Supplemental Fig. S1g. In Fig. 1k–n and Supplemental Figs. S1h, we present additional examples of cerebrovascular amylin-Aβ co-localization from IHC analyses in a subset of brains of patients with fAD and with documented amylin accumulation through IHC and ELISA[21]. Aβ deposits are present in perivascular spaces and arterial walls, whereas amylin appears to accumulate within the lumen (Fig. 1k, l), arterial wall (Fig. 1m) and on the luminal side (Fig. 1n). The IHC analysis detected amylin in approximately 2/3 of the total blood vessels staining positive for Aβ in fAD brains (Table 3). Confocal microscopic analysis of brain section triple stained with anti-amylin, anti-Aβ, and anti-α smooth muscle cell (SMC) actin antibodies further supports co-localization patterns in which Aβ is present in perivascular areas and amylin within the blood vessel wall (Supplemental Fig. S1i).

Our data show accumulation of pancreatic amyloid-forming amylin in the blood and circulating monocytes, and amylin co-localized with Aβ in the brain vasculature, in persons with AD. The results suggest the hypothesis that increased concentrations

**Table 2 Clinical and neuropathological information, age and sex of each individual included in the plasma amylin and brain amylin and Aβ analyses by ELISA, and in the IHC analysis of cerebrovascular amylin-Aβ co-localization.**

| # | Sex | Age | Type-2 DM | Braak stage | Amyloid Angiopathy | MMSE | CERAD |
|---|-----|-----|-----------|-------------|--------------------|----- |-------|
| 1 | F | 95 | NO | 0 | NO | 26 | A=CERAD possible |
| 2 | F | 97 | NO | 4 | NO | 24 | A=CERAD possible |
| 3 | M | 92 | NO | 0 | NO | 30 | NO |
| 4[a] | F | 90 | NO | 3 | NO | 29 | NO |
| 5 | F | 75 | NO | 1 | NO | 23 | NO |
| 6 | F | 84 | NO | 0 | NO | 30 | NO |
| 7 | F | 84 | NO | 1 | NO | 29 | NO |
| 8[a] | F | 85 | NO | 4 | MILD | 23 | NO |
| 9 | F | 93 | NO | 2 | NO | 30 | NO |
| 10 | F | 96 | NO | 2 | MODERATE | 30 | NO |
| 11[a] | F | 90 | NO | 2 | MILD | 30 | NO |
| 12[a] | F | 80 | NO | 1 | NO | 28 | NO |
| 13 | M | 87 | YES | 2 | NO | 27 | NO |
| 14[a] | F | 84 | YES | 1 | MILD | 26 | NO |
| 15 | F | 92 | YES | 2 | NO | 29 | NO |
| 16[a] | F | 87 | YES | 3 | NO | 28 | NO |
| 17[a] | F | 90 | YES | 2 | NO | 29 | NO |
| 18[a] | F | 69 | YES | 1 | NO | 29 | NO |
| 19 | F | 87 | NO | 4 | MILD | 30 | C=Definite AD |
| 20 | F | 78 | NO | 6 | MILD | 0 | C=Definite AD |
| 21[a] | F | 91 | NO | 2 | MODERATE | 29 | B=CERAD probable |
| 22 | M | 78 | NO | 6 | MILD | 15 | C=Definite AD |
| 23[a] | M | 85 | NO | 6 | MODERATE | 16 | B=CERAD probable |
| 24[a] | F | 93 | NO | 2 | NO | 29 | B=CERAD probable |
| 25[a] | M | 84 | NO | 4 | MODERATE | 24 | B=CERAD probable |
| 26 | M | 89 | NO | 1 | MILD | 26 | C=Definite AD |
| 27 | M | 73 | NO | 6 | NO | 16 | C=Definite AD |
| 28 | F | 88 | NO | 3 | NO | 28 | C=Definite AD |
| 29[a] | M | 75 | NO | 6 | SEVERE | 5 | C=Definite AD |
| 30 | F | 101 | NO | 4 | NO | 26 | B=CERAD probable |
| 31[a] | M | 98 | NO | 2 | SEVERE | 24 | B=CERAD probable |
| 32[a] | F | 92 | NO | 5 | MILD | 19 | C=Definite AD |
| 33 | M | 79 | NO | 6 | MILD | 6 | C=Definite AD |
| 34[a] | F | 91 | NO | 5 | MILD | 13 | C=Definite AD |
| 35 | M | 67 | NO | 6 | MILD | 11 | C=Definite AD |
| 36[a] | F | 87 | NO | 3 | MILD | 30 | B=CERAD probable |
| 37 | F | 98 | NO | 4 | NO | 13 | B=CERAD probable |
| 38 | M | 91 | NO | 1 | SEVERE | 28 | NO |
| 39 | F | 79 | YES | 6 | SEVERE | 19 | C=Definite AD |
| 40 | F | 86 | YES | 2 | NO | 30 | B=CERAD probable |
| 41 | F | 91 | YES | 3 | MILD | 25 | B=CERAD probable |
| 42 | F | 91 | YES | 5 | MODERATE | 9 | C=Definite AD |
| 43 | M | 87 | YES | 3 | NO | 21 | A=CERAD possible |
| 44 | M | 75 | YES | 5 | MILD | 28 | C-Definite AD |
| 45 | F | 86 | YES | 6 | MILD | 14 | C-Definite AD |
| 46[a] | M | 77 | YES | 4 | MILD | 30 | B=CERAD probable |
| 47[a] | M | 86 | YES | 2 | NO | 28 | A=CERAD possible |
| 48[a] | M | 81 | YES | 1 | NO | 26 | A=CERAD possible |
| 49 | M | 86 | YES | 6 | MODRATE | 29 | C-Definite AD |
| 50 | F | 93 | YES | 4 | MODERATE | 28 | B=CERAD probable |
| 51 | M | 96 | YES | 4 | MILD | 21 | B=CERAD probable |
| 52 | M | 89 | YES | 4 | MODERATE | 26 | B=CERAD probable |
| 53 | M | 85 | YES | 6 | SEVERE | 4 | C-Definite AD |
| 54 | F | 83 | YES | 4 | MODERATE | 23 | B=CERAD probable |
| 55 | F | 81 | YES | 5 | MILD | 14 | C-Definite AD |
| 56 | F | 85 | YES | 2 | NO | 24 | B=CERAD probable |
| 57 | F | 84 | YES | 6 | SEVERE | 7 | C-Definite AD |
| 58 | F | 81 | YES | 5 | MODERATE | 13 | C-Definite AD |
| 59 | F | 88 | YES | 6 | MODERATE | 29 | C-Definite AD |
| 60 | M | 85 | YES | 3 | NO | 30 | A=CERAD possible |
| **#** | **Sex** | **Age** | **Type-2 Diabetes** | **AD** | **Amyloid Angiopathy** | | **Clinical finding** |
| 1[b] | M | 77 | YES | No | NO | | MCI |
| 2[b] | M | 82 | YES | No | Severe | | MCI |
| 3[b] | M | 83 | YES | Yes | Severe | | DEMENTED |

**Table 2 (continued)**

| # | Sex | Age | Type-2 DM | Braak stage | Amyloid Angiopathy | MMSE | CERAD |
|---|---|---|---|---|---|---|---|
| 4[b] | F | 77 | YES | No | NO | | DEMENTED |
| 5[b] | M | 90 | NO | Yes | Severe | | DEMENTED |
| 6[b] | F | 91 | NO | No | Severe | | NORMAL |
| 7[b] | M | 93 | NO | Yes | Severe | | DEMENTED |
| 8[b] | F | 77 | YES | No | NO | | NORMAL |
| 9[b] | M | 88 | NO | No | Severe | | DEMENTED |
| 10[b] | M | 91 | NO | Yes | Severe | | DEMENTED |
| 11[b] | M | 84 | NO | No | NO | | NORMAL |
| 12[b] | M | 95 | NO | No | NO | | MCI |
| 13[b] | F | 81 | YES | No | NO | | NORMAL |
| 14[b] | F | 90 | NO | Yes | Severe | | DEMENTED |
| 15[b] | F | 87 | YES | Yes | NO | | DEMENTED |
| 16[b] | M | 87 | NO | Yes | Severe | | DEMENTED |
| 17[b] | F | 77 | YES | Yes | Severe | | DEMENTED |
| 18[b] | M | 79 | YES | yes | Severe | | DEMENTED |

The absence/presence of diabetes was determined during life (at longitudinal clinical visits) by patient or caregiver self-report and the use of diabetic medications. The assessment of clinical dementia and the neuropathologic features - neuritic amyloid plaques (Consortium to Establish a Registry for Alzheimer's Disease; CERAD), Braak NFT stage and cerebral amyloid angiopathy (CAA) severity - were scored as previously described[27–31]. [a]Plasma samples from these patients were used in amylin ELISA measurements. [b]These brain samples were used in immunohistochemistry.

**Table 3 Summary of histopathological analysis of cerebrovascular amylin-Aβ deposition in sAD and fAD brains.**

| Cerebrovascular amylin, Aβ and amylin-Aβ deposits (sAD) | Patients/total (%) |
|---|---|
| Independent amylin deposits in vasculature | 09/18 (50%) |
| Independent Aβ deposits in vasculature | 07/18 (38.89%) |
| Mixed amylin-Aβ deposits in vasculature | 09/18 (50%) |
| **Cerebrovascular amylin, Aβ and amylin-Aβ deposits (fAD)** | **Patients/total (%)** |
| Independent amylin deposits in vasculature | 23/32 (71%) |
| Independent Aβ deposits in vasculature | 02/32 (6.25%) |
| Mixed amylin-Aβ deposits in vasculature | 28/32 (87.5%) |

of amyloid-forming amylin in the blood perturb cerebral Aβ efflux, possibly involving inflammation.

**In rats, pancreatic expression of human amylin induces inflammation and cerebrovascular amylin-Aβ deposition (amylin vasculopathy) leading to late-life onset behavior deficits.** We used rats in which pancreatic β-cells express human amylin (the HIP rat model[18]) and wild type (WT) rats expressing the endogenous non-amyloidogenic rat amylin, to determine potentially causative effects of increased concentrations of amyloid-forming human amylin in the blood on the development of systemic and cerebrovascular inflammation. The specificity of human *amylin* RNA expression in the pancreas and lack of human *amylin* RNA in the brain in HIP rats was documented by qRT-PCR, as reported previously[25]. Male and female HIP rats develop type-2 DM associated with pancreatic amylin amyloid deposition[18,32] glucose dysregulation[18,32] and neurological deficits[18,25]. Glucose dysregulation and behavioral deficits develop later (by ~6 months) in HIP female vs. male rats, as we reported previously[18]. Blood amylin concentrations increase with higher blood glucose (Fig. 2a). At the age at which HIP rats develop neurological deficits (~16-months) (Supplemental Fig. S2), blood amylin concentrations in HIP male rats were within a similar range as in persons with cognitive decline (Fig. 2b).

Using flow cytometry, we found higher numbers of CD14+ monocytes and amylin-positive CD14+ monocytes in blood from HIP rats compared to those in the blood from WT littermates (Fig. 2c). As in human blood from persons with AD (Fig. 1f), confocal microscopic imaging of circulating monocytes stained for CD14+ (have also amylin deposits (Fig. 2d).

Pancreatic amyloid-forming amylin activates the NLRP3 inflammasome in macrophages and dendritic cells leading to interleukin (IL)-1β maturation and pancreatic islet inflammation[11–13]. The concentration of pro-inflammatory cytokine IL-1β in plasma was higher in HIP rats than in WT littermates (Fig. 2e). This was associated with increased IL-1β immunoreactivity signal intensity in HIP rat brain blood vessel walls (Fig. 2f), consistent with previously published data from IL-1β analysis in HIP rat brain parenchymal tissue[25,26]. Using IHC with anti-amylin and anti-glial fibrillar acidic protein (GFAP) antibodies, we detected cerebrovascular amylin-induced astroglial reaction in HIP rat brains (Fig. 2g). This appears to be associated with vascular recruitment of monocytes and macrophages as suggested by IHC with antibodies against Cluster of Differentiation (CD) 68 and CD11b (Fig. 2h, i). These results indicate chronic exposure to amyloid-forming amylin secreted from the pancreas as a trigger of systemic- and local cerebrovascular inflammation.

The Thioflavin T (ThT) fluorescence signal intensity in blood lysates was higher in HIP rats than in WT littermates (Fig. 3a) indicating amyloid-forming amylin accumulation in the blood. Using ELISA, we found increased amylin concentrations in HIP rat brain microvessel lysates compared to brain microvessel lysates from WT littermates, age 16-months (Fig. 3b). Co-staining of brain slices from HIP and WT rats with anti-Aβ and anti-amylin antibodies detected vascular amylin-Aβ deposition in HIP rats (Fig. 3c). In HIP rat brains, amylin immunoreactivity was detected on the luminal side of the blood vessel, often with Aβ within perivascular spaces and scattered through parenchymal tissue.

We further employed APP/PS1 rats with pancreatic expression of human amylin (APP/PS1/HIP rats) to study the impact of pancreatic amyloid-forming human amylin on cerebrovascular Aβ in the setting of AD-like pathology. APP/PS1/HIP rats develop amylin-Aβ deposits in the brain[21]. Compared to APP/PS1 littermates, APP/PS1/HIP rats have increased Th-T signal intensity in blood lysates (Fig. 3d) and amylin accumulation in brain microvessels (Fig. 3e). IHC with anti-Aβ (green) and anti-amylin (brown) antibodies revealed vascular tissue areas of amylin-Aβ co-localization in APP/PS1/HIP rat brains, whereas APP/PS1 littermates had no cerebrovascular amylin-Aβ deposits (Fig. 3f).

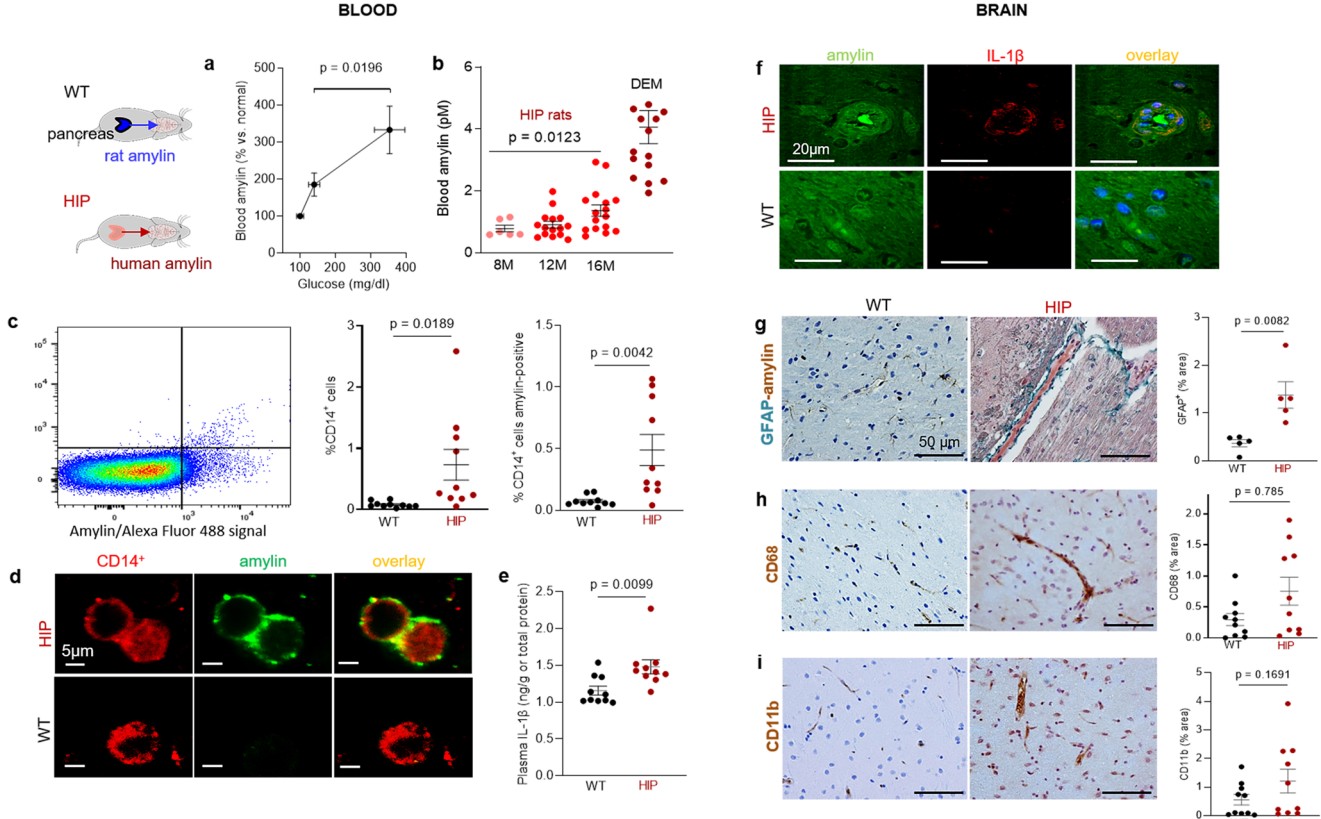

**Fig. 2 Systemic and cerebrovascular inflammation in rats with pancreatic expression of amyloid-forming human amylin (HIP rats). a** Cross-sectional blood amylin and glucose concentrations in HIP rats age 6–8 months (n = 6), age 10–12 months (n = 13), and age 15–16 months (n = 12). **b** Blood amylin concentrations in humans with dementia (DEM) vs. HIP rats; same rats as in (n = 16) (**a**). **c** Flow cytometry sorting of circulating CD14+ monocytes positive for amylin in blood from same rats as in (**b**) (n = 10 males/group). **d** Confocal microscopic images of circulating monocytes stained for CD14+ (red) and amylin (green) in blood from the same rats as in (**b**) (n = 5 blood samples/group). **e** Interleukin (IL)-1β ELISA in plasma from HIP vs WT rats similar to groups in (**b**) (n = 10 males/group). **f** Confocal microscopic images showing IL-1β and amylin deposits in brain blood vessels in rats studied in (**b**) (n = 3 males/group). **g** IHC analysis of brain sections from HIP and WT rats from the same groups as in (**c**) showing vascular deposits of amylin (brown) and astroglial reactions (green stains for glial fibrillar acidic protein; GFAP) (n = 5 males/group). IHC analysis of phagocytic microglia (CD68) (**h**) and vascular monocyte recruitment (CD11b) (**i**) in brain sections from HIP vs WT rats from the same groups as in (**b**) (n = 10 males/group). Data are means ± SEM; unpaired t-test for all panels.

This was associated with vascular recruitment of monocytes and macrophages as indicated by IHC with antibodies against CD68 and CD11b (Fig. 3g, h), similar to cerebrovascular inflammation demonstrated in HIP rats (Fig. 2h, i).

Our results show that late-life onset of neurological deficits in rats with pancreatic-specific expression of human amylin (Supplemental Fig. 2) are associated with amylin deposition in cerebral arterioles (including co-deposits with Aβ) (Fig. 3c, f) and in capillaries (Fig. 2b, e), and with the development of systemic and cerebrovascular inflammation (Figs. 2 and 3g, h). These results replicate our findings in humans (Fig. 1).

**Amylin vasculopathy impairs cerebral Aβ efflux through altering cerebral vasodilation.** Impaired interstitial fluid drainage in the brain is indicated by the presence of perivascular Aβ deposits (as in Fig. 3c, f) and is attributed to alterations in contractility and relaxation of vascular SMCs[2–4]. Endothelial nitric oxide (NO)-arginase homeostasis, a mediator of vascular myogenic tone[33], is impaired in HIP rats and is associated with endothelial dysfunction[34]. We used cerebral blood flow (CBF), pressure myography, and vascular SMC oxidative stress experiments in HIP versus WT rats to test a possible association between increased concentrations of amyloid-forming amylin in the blood and impairment of cerebral vasodilation.

Longitudinal brain MRI measurements revealed consistent structural alterations that progressed more rapidly with aging in HIP vs. WT rats (Fig. 4a). Cerebral perfusion was assessed using pseudo-continuous arterial spin labeling (ASL)[35]. The results show reduced CBF in HIP rats, age 15–16 months (Fig. 4b). Plasma concentrations of nitrite and nitrate (stable NO end products) were increased in HIP vs. WT rats, age 15–16 months (Fig. 4c), possibly as a result of systemic inflammation in HIP rats (as suggested by data in Fig. 2). Pressure myography experiments using isolated pial arteries demonstrate that both WT and HIP arteries develop arterial tone[36] (e.g., pressure-induced constriction) (Fig. 4d); however, arteries from HIP rats show elevated arterial tone compared to WT rats with increasing intravascular pressure (e.g., 60–100 mmHg) (Fig. 4d, e). To further verify the amylin-induced impairment of arterial tone, we used cerebral arteries isolated from WT rats and amylin knockout (AKO) rats, and from AKO rats intravenously injected with amyloid-forming human amylin at a regimen that replicates the amylin immunoreactivity signal intensity measured in human plasma[18] (i.e., 60 μg/kg body weight; daily IV injection via tail vein for 1 week). Arterial tone is similar in cerebral arteries from WT and AKO rats (Fig. 4f); however, arterial tone was elevated in arteries from rats injected with human amylin (Fig. 4f). These results suggest a direct effect of amylin on impairing arterial tone. Consistent with these results, vascular SMCs from

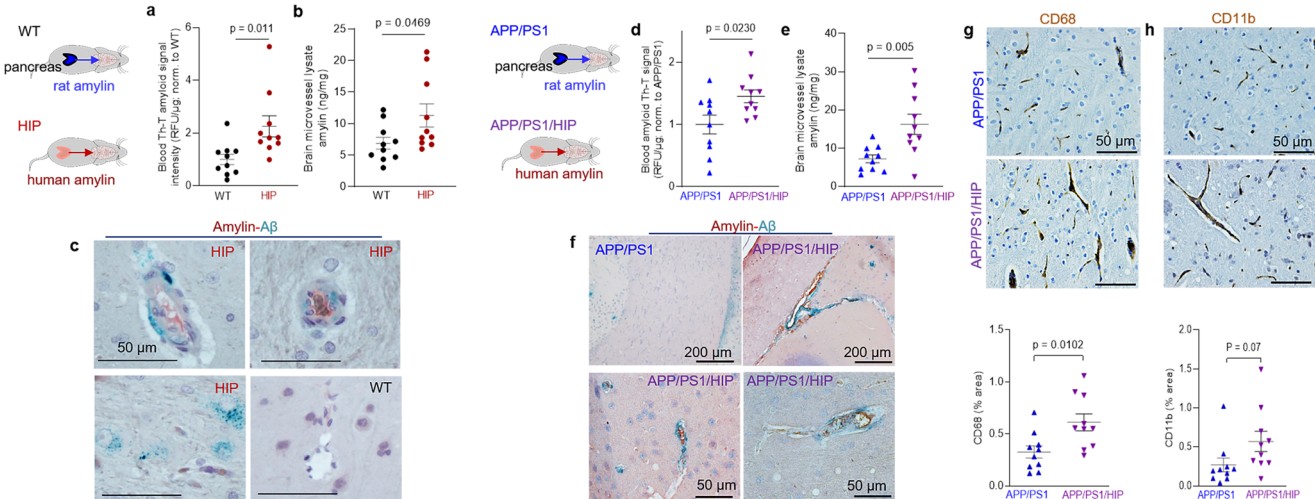

**Fig. 3 Cerebrovascular amylin-Aβ deposition induced by amyloid-forming human amylin secreted from the pancreas. a** Thioflavin T (Th T) fluorescence signal intensities in blood lysates from HIP rats and WT littermates (age 15-16-months; $n = 10$ males/group). **b** Amylin concentrations in brain microvessel lysates in HIP and WT rats similar to those in (**a**) ($n = 10$ males/group). **c** IHC analysis of HIP rat brains showing Aβ deposits (green) in perivascular spaces and amylin accumulation (brown) within the lumen. ($n = 5$ males/group; age 15-16-months). **d** Average Thioflavin T (Th T) fluorescence signal intensities in blood lysates from APP/PS1/HIP and APP/PS1 littermates age 15-16-months ($n = 10$ rat males/group). **e** Amylin concentrations in brain microvessel lysates from same rats as above. **f** Representative IHC micrographs of brain sections from APP/PS1/HIP and APP/PS1 rats co-stained with anti-amylin (brown) and anti-Aβ (green) antibodies ($n = 5$ males/group; age 15-16-months) (3 slides/brain). Representative IHC images and analysis of phagocytic microglia (CD68) (**g**) and vascular monocyte recruitment (CD11b) (**h**) in brain sections from APP/PS1/HIP vs APP/PS1 rat males ($n = 10$ males/group; age 16-months). Data are means ± SEM; unpaired $t$-test for all panels.

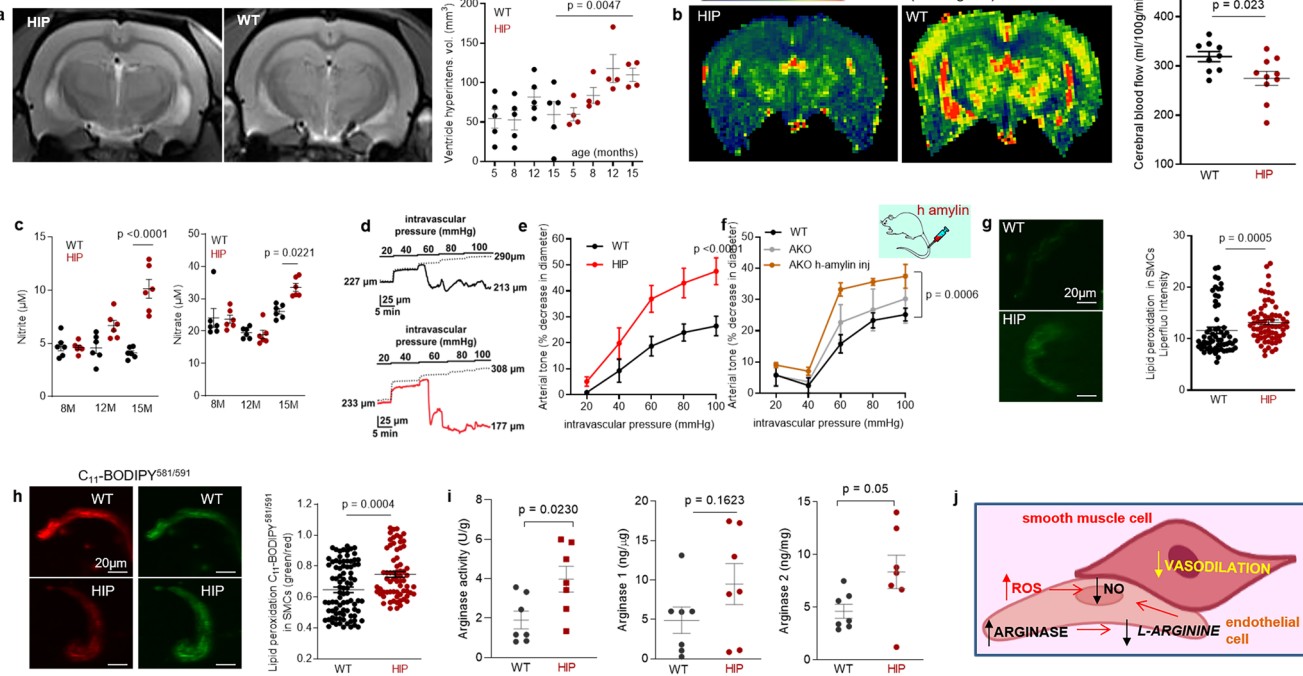

**Fig. 4 Altered relaxation of vascular smooth muscle cells by increased blood amylin concentrations. a** T$_2$-weighted MRI and longitudinal ventricular hyperintensity volumes in HIP vs. WT rats ($n = 4$–5 males/group). **b** CBF maps and global CBF in HIP and WT rats, age 15-16 months (9 males/group). **c** Cross-sectional concentrations of plasma nitrite and nitrate in HIP and WT rats ($n = 6$ males/group/age). Diameter traces in pial arteries from HIP and WT rat males at different intravascular pressures (**d**), and arterial tone of pial arteries (**e**) measured at the indicated intravascular pressure (2–3 arteries/rat, $n = 6$–7 males/group, age 15–16 months). **f** Same as in (**e**) in posterior cerebral arteries from WT and amylin knockout (AKO) rats, and in AKO rats intravenously injected with human amylin ($n = 3$ males/group, age 9–10 months). **g, h** Lipid peroxidation in pial artery SMCs from WT and diabetic HIP rat males measured with Liperfluo (**g**; $N = 62$ SMC from 4 WT rats and 70 cells from 4 HIP rats) and C$_{11}$-BODIPY$^{581/591}$ (**h**; $N = 87$ SMC from 7 WT rats and 68 cells from 4 HIP rats). **i** Arginase activity and arginase-1 and arginase-2 concentrations in HIP vs. WT brain microvessel lysates ($n = 7$ males/group; age 15–16 months). **j** Proposed mechanism: chronically increased concentrations of pancreatic amyloid-forming amylin in blood cause oxidative stress within the vascular wall leading to NO-arginase dysregulation and impaired SMC function and myogenic tone. Data are means ± SEM. Mann–Whitney non-parametric test for panels (**g, h**), unpaired $t$-test for the other panels.

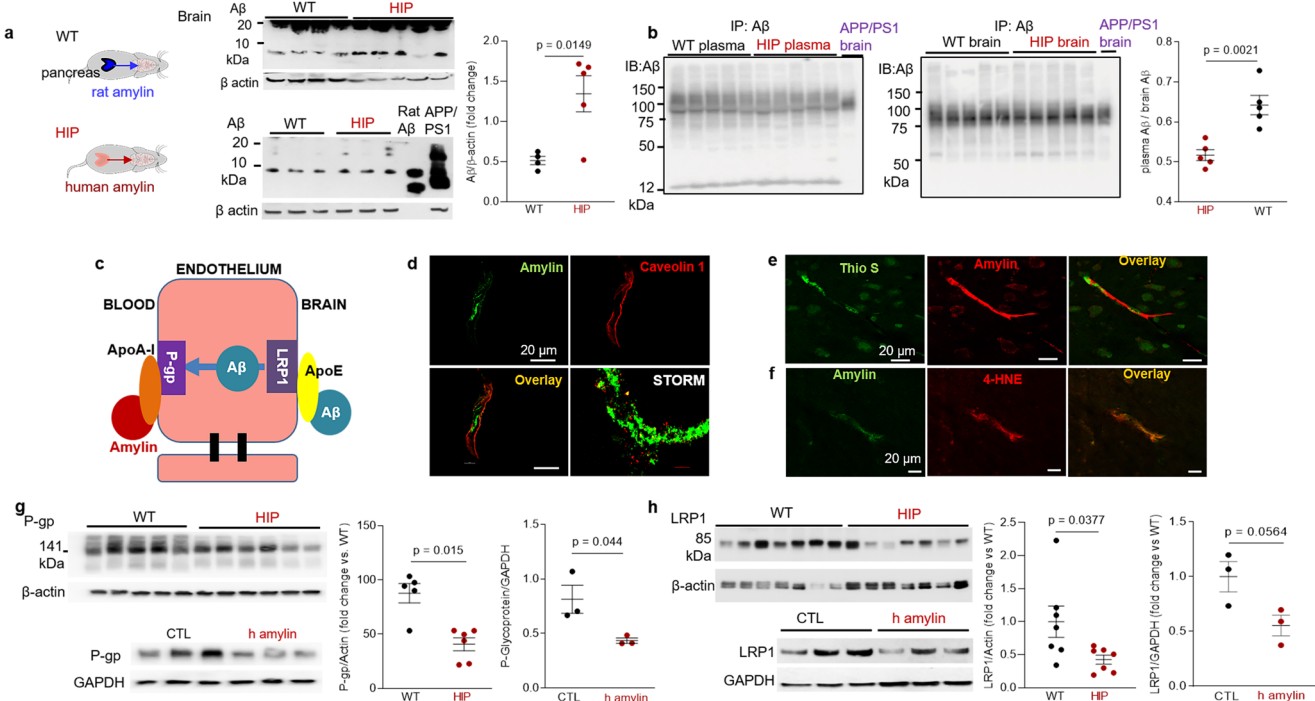

**Fig. 5 Impaired Aβ efflux from the brain induced by amyloid-forming human amylin secreted from the pancreas. a** Western blot analyses of brain tissue Aβ in WT and HIP rats ($n = 4–5$ males/group; age 15–16-months) with rat Aβ$_{40}$ and APP/PS1 rat brain homogenate used as positive controls for Aβ immunoreactivity signal. **b** Estimated brain Aβ efflux through immunoprecipitation and Western blot analyses of Aβ in plasma and brain homogenates from WT and HIP rats (similar to those in **a**), with APP/PS1 rat brain homogenate used as positive control for Aβ immunoreactivity signal. **c** Proposed amylin action on the Aβ transport across the BBB. **d** Confocal microscopy and STORM analysis of amylin in brain microvessels isolated from HIP ($n = 47$ microvessels) and WT ($n = 21$ microvessels) rats ($n = 3$ males/group; age 15–16 months). Confocal microscopy analysis of serial sections from HIP rat brains stained with Thioflavin S (Thio S) or amylin (**e**) and with the lipid peroxidation marker 4-HNE or amylin (**f**) ($n = 3$ males similar to those in **d**). Western blot analysis of P-gp (**g**) and LRP1 (**h**) in brain microvessel lysates from HIP and WT rats ($n = 5–7$ males/group similar to those in **a**), and in brain microvascular ECs incubated with human amylin. Data are means ± SEM.

HIP rats have increased lipid peroxidation (Fig. 4g, h), which also occurs in SMCs incubated with exogenous human amylin (Supplemental Fig. S3a, b).

Increased lipid peroxidation contributes to oxidative stress in the vascular wall and reduces NO bioavailability, altering vasodilatory function. Both arginase activity and expression were increased in brain microvascular lysates from HIP rats compared to those from WT littermates (Fig. 4i), suggesting arginase-NO dysregulation (Fig. 4j; proposed mechanism). A possible impact of increased blood amylin concentration on cerebrovascular arginase-NO regulation was further tested in brain microvascular lysates from rats intravenously injected with amyloid-forming human amylin (Supplemental Fig. S3c).

Taken together, our results indicate perivascular Aβ deposits in HIP and APP/PS1/HIP rats (Fig. 3c, f) are potentially linked to altered spontaneous contraction/relaxation of cerebrovascular SMCs caused by the development of amylin vasculopathy.

**Amylin deposition in brain capillaries suppresses P-gp and LRP1 protein expression.** Western blot analyses of Aβ in brain tissue homogenates (Fig. 5a) and Western blot analysis of enriched Aβ by immunoprecipitation in plasma and brain homogenates (Fig. 5b) show brain Aβ deposition in HIP rats, even in the absence of genetically induced Aβ overexpression (i.e., as in APP/PS1 rats). The ratio of plasma-to-brain Aβ levels was lower in HIP compared to WT rats suggesting potential impairment Aβ transport from brain to blood. In these experiments, we used brain homogenates from a 12-month-old APP/PS1 rat as the positive controls for brain Aβ accumulation.

Aβ transport across the BBB is mediated by the low-density lipoprotein receptor-related protein 1 (LRP1), an apolipoprotein E (APOE) receptor[37–39]. At the BBB, LRP1 binds Aβ on the brain side of the endothelium facilitating Aβ release into the systemic circulation (Fig. 5c). P-glycoprotein 1 (P-gp; also known as ATP-binding cassette sub-family B member 1; ABCB1), an ATP-dependent efflux pump, mediates the release of Aβ at the blood side of the BBB[40,41]. ApoA-I stabilizes P-gp in endothelial cells (ECs)[42,43] and binds to amylin in HIP rats, as we described in our previous study[18]. Here, we measured associations of cerebrovascular amylin deposition with EC stress-mediated alterations in LRP1 and P-gp protein expression in the HIP rat brain microvasculature. Confocal microscopy and analysis of amylin deposits using Stochastic Optical Reconstruction Microscopy (STORM) super-resolution imaging showed the juxtaposition of amylin and caveolin-1 optical intensity signals (Fig. 5d), confirming amylin deposition in HIP rat brain capillaries (see Fig. 3b). Amylin deposition in brain capillaries has biochemical properties of amyloid (Fig. 5e) and triggers accumulation of the lipid peroxidation marker 4-hydroxynonenal (4-HNE) (Fig. 5f) demonstrating amylin amyloid-induced oxidative stress within the BBB. This was associated with downregulated P-gp and LRP1 protein levels, as indicated by Western blot analysis of these proteins in brain capillary lysates from HIP and WT rats and in lysates of ECs incubated with amyloid-forming human amylin (Fig. 5g, h).

*In vitro BBB model of Aβ transcytosis with amylin-mediated EC endothelial stress.* To determine whether amyloid-forming amylin may directly influence Aβ efflux as mediated by LRP1, we employed an in vitro model of BBB in which the EC monolayer

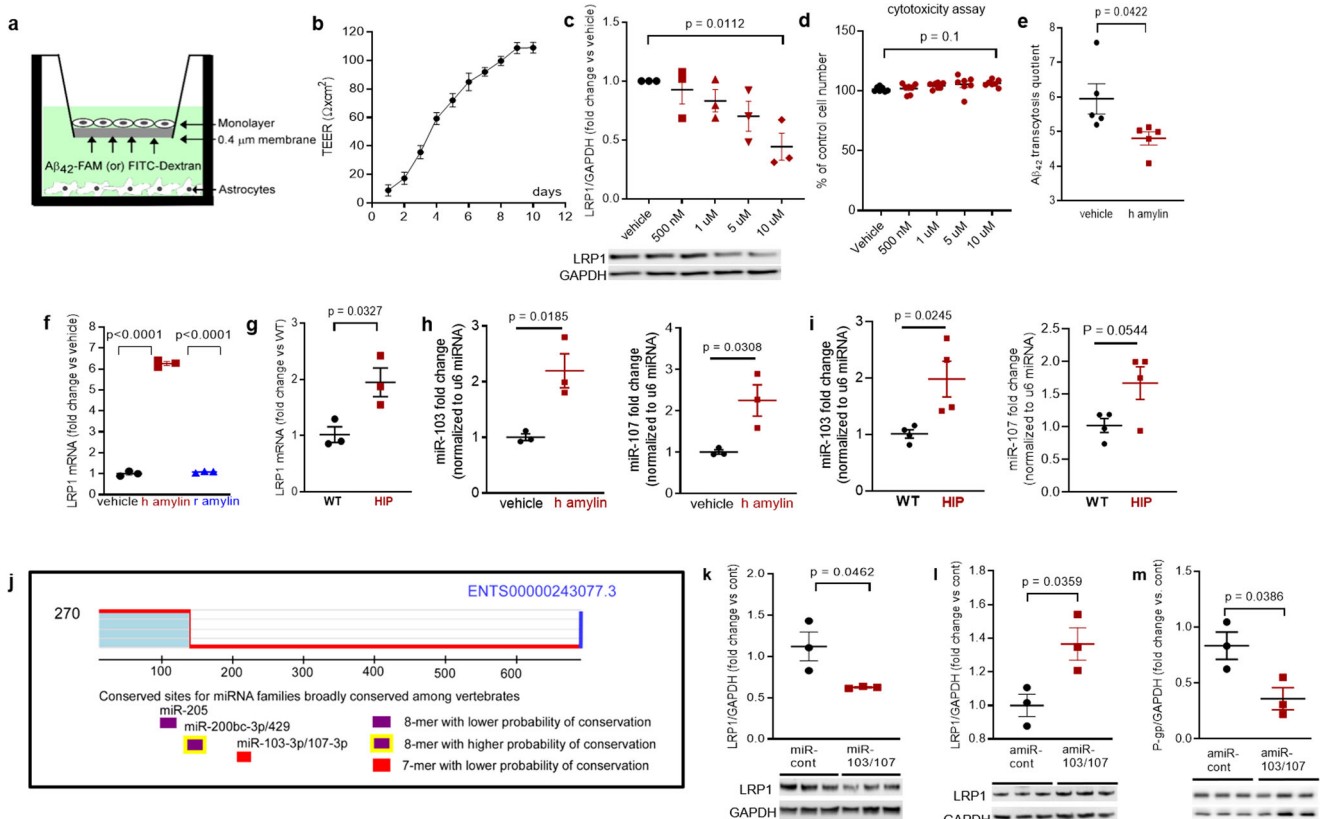

**Fig. 6 In vitro test of amylin-induced impairment of Aβ efflux across the BBB. a** Cartoon representation of the in vitro BBB model (ECs monolayer - luminal chamber; astrocytes - abluminal chamber) used in Aβ transcytosis experiments. **b** Transendothelial electrical resistance (TEER) in EC monolayers ($n = 20$ preparations) as a function of days in culture. **c** Representative Western blot and densitometry quantification of LRP1 in lysates from primary rat brain microvascular vascular ECs treated with vehicle or various concentrations of human amylin (500 nM, 1 μM, 5 μM, and 10 μM) for 24 h ($n = 3$ preparations/test). **d** Percent cell viability from the MTS assay in ECs treated with amyloid-forming human amylin (500 nM, 1 μM, 5 μM, and 10 μM) or vehicle, for 24 h. **e** The Aβ$_{42}$ transcytosis quotient (TQ) across the in vitro BBB, in vehicle- and human amylin-treated EC monolayers. **f** LRP1 mRNA levels (fold difference using $2^{-\Delta\Delta Ct}$ method) measured with qRT-PCR in lysates from ECs treated with vehicle, human amylin or rat amylin. **g** LRP1 mRNA levels measured by qRT-PCR in brain capillary lysates from same rats as in Fig. 5h. **h, i** miRNA (miR)-103 and miR-107 expression levels measured by qRT-PCR in lysates from ECs treated with vehicle or human amylin (same as in Fig. 5h), and in brain capillary lysates from same rats as in Fig. 5h. **j** TargetScan schematic showing consensus regions for miR-205, miR200bc-3p/429, and miR-103 and miR-107. Western blot analyses of LRP1 from miRNA (miR) 103 and miR-107 treated ECs compared to miR-control ($n = 3$ preparations/group) (**k**), as well as of LRP1 (**l**) and P-gp (**m**) from antagomir (amiR) 103 and amiR-107 treated ECs compared to amiR-control treated cells ($n = 3$ preparations/group). Data are mean ± SEM. one-way ANOVA with Dunnett's post hoc (F). unpaired t test for the other panels.

was exposed to amyloid-forming human amylin on the luminal side and Aβ at the abluminal (brain side), as shown in Fig. 6a. The BBB model was tested for monolayer formation by measuring trans-endothelial electrical resistance (TEER) (Fig. 6b). All experiments were performed with a fully formed EC monolayer characterized by the maximum $TEER = 110 \pm 5$ Ω/cm². The dose response of brain microvascular ECs to incubation with various concentrations of human amylin for 24 h in shown in Fig. 6c. LRP1 protein levels decreased with increasing concentrations of human amylin; LRP1 expression was reduced by more than 50% in ECs incubated with 10 μM human amylin (Fig. 6c). EC viability was not affected by incubation with human amylin (Fig. 6d), indicating that decreased LRP1 protein expression is not due to cell death. Next, we applied human amylin peptide (10 μM) or vehicle at the luminal (blood) side for 24 h followed by washing the EC monolayers with PBS and application of FITC-Dextran or carboxyfluorescein labeled Aβ$_{42}$ (Aβ$_{42}$-FAM) at abluminal (brain) side of the BBB for 1-hour. The amounts of Aβ$_{42}$-FAM and FITC-Dextran that crossed the monolayer were estimated from the fluorescence intensity in the medium samples collected from the luminal side and used to calculate the Aβ transcytosis quotient. Amylin-pretreatment reduced the Aβ transcytosis quotient by $20 \pm 5\%$ ($P < 0.05$) (Fig. 6e).

In ECs incubated with 10 μM amyloid-forming human amylin for 24 h, the LRP1 mRNA levels were elevated compared to control ECs (vehicle treated) and ECs incubated with the same concentration of rat amylin (Fig. 6f). LRP1 mRNA levels were increased in HIP vs. WT brain capillary lysates (Fig. 6g). Taken together, the results indicate that amyloid-forming amylin may directly influence Aβ efflux through suppressing transport protein expression at a post-transcriptional level.

*Antisense microRNAs 103/17 rescue amylin-induced LRP1 suppression.* Published data show that paralog miRNAs *miR-103* and *miR-107* are upregulated by oxidative stress[44] and repress LRP1 translation in several cell lines[45]. We, therefore, hypothesized that *miR-103* and *miR-107* are involved in amylin-induced LRP1 downregulation in the microvascular ECs. ECs incubated with 10 μM amyloid-forming human amylin for 24 h showed increased *miR-103* and *miR-107* levels, as indicated by the RT-PCR data from EC lysates (Fig. 6h). Using the same HIP and WT rat capillary lysates as in above (Fig. 6g), we detected higher

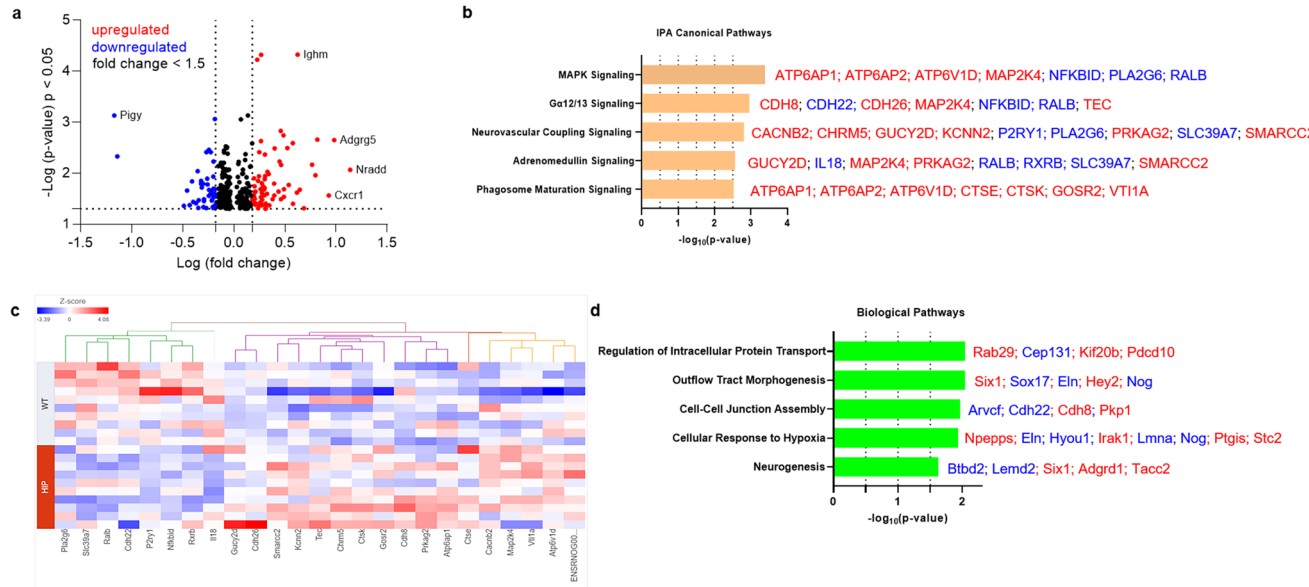

**Fig. 7 The genes predicted to influence the impact of cerebrovascular amylin deposition on the central nervous system. a** The $\log_{10}$ (p-value) versus $\log_{10}$ (fold change) of the DE genes in HIP vs. WT rat brains. Each dot represents one gene, with upregulated in red color (HIP vs. WT, $\geq$ 1.5-fold change), downregulated in blue color (HIP vs. WT, $\geq$ 1.5-fold change), and < 1.5-fold change in black color. **b** Top 5 canonical pathways identified by Ingenuity Pathways Analysis of differentially expressed (DE) genes detected by RNA-seq analysis of hippocampal tissue from the HIP vs. WT rat groups ($P < 0.05$) ($n = 10$ males/group). **c** Hierarchical clustering of 25 DE genes identified the enrichment of top 5 canonical pathways in (**b**). **d** Top 5 Gene Ontology (GO) biological processes enriched in HIP rat group compared to WT rat group.

miR-103 and miR-107 levels in HIP vs. WT rat capillaries (Fig. 6i), consistent with the in vitro results (Fig. 6h). TargetScan predicts that miR-103 and miR-107 bind directly to LRP1 (Fig. 6j). Therefore, we co-transfected miR-103 and miR-107 mimics into microvascular ECs, and then used antagomir (amiR)-103 and amiR-107 to silence amylin-induced upregulation of miR-103 and miR-107. Our results show miR-103/107 downregulate LRP1 (Fig. 6k), replicating the effect of the incubation with amyloid forming human amylin (Fig. 5h). AmiR-103/107 rescue LRP1 expression in ECs following amylin-induced cell stress (co-incubation with 10 μM human amylin) (Fig. 6l); however, amiR-103/107 does not rescue P-gp expression (Fig. 6m).

Taken together, our results (Figs. 2–6) suggest a potential link between chronically dysregulated pancreatic amylin and impaired rat Aβ efflux from the brain into the systemic circulation through mechanisms that involve: 1, amylin vasculopathy leading to reduced cerebral vasodilation and altered interstitial fluid drainage along the walls of cerebral blood vessels; and 2, amylin-induced endothelial dysfunction leading to P-gp and LRP1 downregulation at the BBB. In HIP rats, causes other than decreased transport, such as altered degradation rates of Aβ, increased aggregation causing immunological hiding of Aβ, enhanced influx of Aβ, or increased blood levels of Aβ could explain the decrease of plasma-to-brain ratio for Aβ.

**Immune, metabolism, and hypoxia-related brain gene expression altered by increased concentrations of amyloid-forming human amylin in the blood.** Previous studies using a TaqMan Gene Expression assay found that both pro-inflammatory and anti-inflammatory microglia genes were differentially expressed (DE) in HIP vs. WT rat groups[25]. To predict possible ample molecular patterns in brain genes associated with chronic exposure to pancreatic amyloid-forming human amylin in the blood, we used RNA-seq and gene expression network analyses of hippocampal genes in HIP vs. WT male rats (age, 15–16-months). Comparison of RNAseq data from HIP vs. WT rats ($n = 10$ males/group) enabled the identification of 408 gene transcripts

that were differentially expressed ($P < 0.05$). The range of characteristic magnitude of change of upregulated and downregulated DE genes is shown in Fig. 7a. DE genes were annotated in the Ingenuity Pathway Analysis (IPA) database, which identified enrichment of multiple canonical pathways represented by these genes (Fig. 7b). Among DE genes in HIP rats compared to WT rats, 25 genes identified the enrichment of the top 5 canonical pathways as shown by the hierarchical clustering heatmap of these DE genes (Fig. 7c). We further categorized the 408 DE genes based on gene ontology (GO) by using the Database for Annotation, Visualization and Integrated Discovery (DAVID) into biological processes enriched in HIP compared to WT rats (Fig. 7d). These results (Fig. 7b–d) suggest dysregulation of genes involved in cellular responses to inflammation, altered metabolism and hypoxia may be associated with chronically increased blood amylin concentrations, cerebrovascular amylin deposition, and impairment of brain Aβ clearance.

## Discussion

Our data from humans and laboratory animal models show that a potentially critical mechanism enabling cerebral Aβ pathology to develop involves cerebrovascular accumulation of amyloid-forming amylin secreted from the pancreas. Because pancreatic amylin contributes to the development of type-2 DM, our results suggest amylin-Aβ interaction as a potential missing molecular link between type-2 DM and AD, and a promising new approach to therapy. We found three interdependent factors that appear to underlie amylin-induced impairment of brain Aβ clearance: 1, concentrations of blood amylin are increased in dementia vs. cognitively unimpaired individuals; 2, chronically increased concentrations of amyloid-forming amylin in the blood promote amylin accumulation in circulating monocytes reflecting systemic inflammation and leading to cerebrovascular amylin deposition; and 3, cerebrovascular amylin deposition perturbs LRP1-Pgp-mediated Aβ transport across the BBB and Aβ clearance through interstitial fluid drainage along vascular walls, as indicated by

amylin-Aβ co-localization in blood vessel walls and perivascular spaces.

Compensatory insulin secretion is central to insulin resistance and coincides with increased amylin secretion[8–10] (also demonstrated by our current data in participants in a cohort spanning the cognitive continuum from cognitively unimpaired to mild cognitive impairment and dementia; Fig. 1d). Given that patients with type-2 DM are insulin resistant for many years before their clinical diagnosis, they are exposed to chronic hyperamylinemia that is linked to pancreatic amylin amyloid deposition[8–10], NLRP3 inflammasome activation and increased secretion of proinflammatory IL-1β from macrophages and dendritic cells[11–13]. Our data showing amylin engulfed by circulating monocytes in association with increased plasma IL-1β concentrations indicate possible innate immune responses to increased concentrations of pancreatic amyloid-forming amylin in the blood. These data suggest that cerebrovascular amylin deposition leading to perivascular inflammation may develop, at least in part, due to an inadequate innate immune reaction to prediabetes-related rising concentrations of pancreatic amyloid-forming in the blood. Future studies should determine whether modulation of the amylin-IL-1β pathway might provide an approach for counteracting neuroinflammation in the setting of AD.

Perivascular Aβ deposits are common in AD brains and have been attributed to impaired interstitial fluid drainage[2–4]. The motive force of interstitial fluid drainage arises from the spontaneous contraction and relaxation of vascular smooth muscle cells[2–4]. Our results show associations between amylin-mediated systemic inflammation and reduced cerebral vasodilation and CBF through arginase-NO dysregulation within the vessel wall. Our data also show that pancreatic amyloid-forming amylin accumulates in brain capillaries and may affect expression of P-gp and LRP1, proteins that mediate Aβ transport across the BBB. Although APOE/LRP1-regulated pathways have a well-established role in brain Aβ clearance[46] the possibility that LRP1 expression can be downregulated by amylin from the luminal side of the blood vessel may represent a novel therapeutic target to reduce AD pathology.

In conclusion, our data suggest that screening for pancreatic amylin dysregulation by measuring amylin accumulation in blood and circulating monocytes could identify people at increased risk for brain microvascular and AD pathologies. This is also important because impaired executive function and working memory are common in persons with type-2 diabetes without dementia[1], and amylin dysregulation is associated with type-2 diabetes and may be a possible contributing factor to these clinical effects. Type-2 DM and AD, two growing global health threats, appear linked through complex mechanisms[14,47–50] possibly involving amyloid-forming amylin as a molecular factor beyond glucose and insulin dysregulation. Amylin-mediated cerebrovascular inflammation, impairment of brain Aβ clearance and dysregulated gene expression in the brain appear as biological characteristics of pancreatic hypersecretion of amylin (prediabetes). The present data may improve understanding of the mechanisms of cerebrovascular inflammation and the risk of impaired Aβ elimination from the brain in the setting of prediabetic insulin resistance, when blood amylin concentration is chronically elevated. Inhibiting systemic and cerebrovascular inflammation caused by pancreatic amyloid-forming amylin could reduce cerebrovascular arginase-NO dysregulation, vasoconstriction, blood flow reductions, and brain Aβ accumulation.

## Methods

**Human samples**. This research employed de-identified whole blood, plasma, and frozen and formalin fixed brain tissue from the biobank of the Alzheimer's Disease Research Center at the University of Kentucky (UK-ADRC) under a protocol approved by the University of Kentucky Institutional Review Board (IRB). Informed consent was obtained prospectively. Whole blood samples were collected from 83 participants in a cohort spanning the cognitive continuum from cognitively unimpaired to mild cognitive impairment and dementia and 80 samples were analyzed (all included on the same amylin ELISA kit). Measurements were performed in duplicate. Frozen temporal cortex tissue samples were obtained from 42 persons with sAD-type dementia documented by amyloid β (Aβ) positivity and 18 cognitively unimpaired individuals. Both plasma and frozen brain tissue were obtained from 20 individuals. Formalin fixed dorsolateral frontal cortex (Brodmann area 9) was obtained from 32 individuals. Summary statistics for individuals providing each type of tissue including sample size, cognitive status, sex, diabetes status and age, along with clinical and neuropathological information, are given in Tables 1, 2. The absence/presence of diabetes was determined during life (at longitudinal clinical visits) by patient or caregiver self-report and the use of diabetic medications. The assessment of clinical dementia and the neuropathological characteristics, including neuritic amyloid plaques, Consortium to Establish a Registry for Alzheimer's Disease (CERAD), Braak NFT stage, and cerebral amyloid angiopathy (CAA) severity, were scored according to published protocols[27–31]. A secondary IHC analysis of cerebrovascular amylin-Aβ deposition was performed in a subset of fAD brains (n = 27) with documented amylin accumulation through IHC and ELISA[21]. Formalin fixed temporal cortex tissues from fAD mutation carriers were provided by the Queen Square Brain Bank for Neurological Disorders at UCL Queen Square Institute of Neurology (United Kingdom) and King's College London (United Kingdom).

Observer-blind analyses were performed on all human tissue samples; the results were communicated to the AD Center at Univ of Kentucky to assess the relationships with AD pathology/cognitive function. Investigators were also blinded in the scoring of histological analyses and in the longitudinal animal behavior testing to prevent bias. Investigators were not blinded during pharmacological intervention/ injection because the interventions were performed by the same personnel. Investigators were not blinded during most of biochemical assays because it is not necessary. Data collection was performed at the same time for experimental groups with the same setting.

**Experimental animals**. This investigation conforms to the Guide for the Care and Use of Laboratory Animals published by the US National Academies Press (8th edition, 2011) and was approved by the Institutional Animal Care and Use Committee at the University of Kentucky. Rats were housed in individually ventilated cages, on a 12 h light cycle, and received a standard pelleted diet and water ad libitum.

We compared rats that develop type-2 diabetes linked to the expression of amyloid-forming human amylin in the pancreas (HIP rat males; n = 84) with control wild type rats that express endogenous non-amyloidogenic rat amylin in the pancreas (WT rats; n = 52). Breeding pairs were purchased from Charles River Laboratory. For cross-sectional blood amylin and glucose analyses, we used HIP rats age 6–8 months, age 10–12 months, and age 15–16 months. For all other experiments, we used rats age 15–16 months. We further employed APP/PS1/HIP rats (n = 10 males; age 15–16 months) to study the impact of pancreatic amyloid-forming human amylin on cerebrovascular Aβ in the setting of AD-like pathology. APP/PS1/HIP rats were generated as previously described[21]. Briefly, TgF344-19 rats from the Rat Resource and Research Center, Univ. of Missouri, Columbia, MO (APP/PS1 rats) are Fischer rats that express human Aβ (A4) precursor protein (hAPP) gene with the Swedish mutation (K595N/M596L), and presenilin 1 (PSEN1) gene with a deletion of exon 9, driven by mouse prion promoter (Prp). The APP/PS1 rats were crossbred with HIP rats to generate rats that are triple transgenic for human amylin, APP, and PSEN1 (APP/PS1/HIP rats). APP/PS1 rats (n = 10 males; age 15–16 months) that express non-amyloidogenic rat amylin served as controls for the amyloidogenic human amylin effects. We also used pancreatic and brain tissues from amylin knockout (AKO)[21] rats (n = 3 males; age 15–16 months) as negative controls for amylin expression. A total of n = 159 rat males was used in the study.

**MRI and CBF**. MRI scans were performed on HIP and WT littermates using a horizontal 7 T nuclear MRI scanner (ClinScan, Brucker BioSpin MRI, Ettlingen, Germany)[18]. Coronal T$_2$-weighted images were obtained using generic parameters: field of view (FOV) 40 mm, repetition time (TR) 3000 ms, echo time (TE) 24 ms, slice thickness 1 mm, inter-slice gap 1 mm, 7 slices. Cerebral perfusion was assessed using the published ASL protocol[35]. The anatomical and perfusion images were co-registered and filtered with the ASL toolbox routines. A mask was manually defined for each rat to exclude both tissue from outside the brain as the thresholding was sometimes insufficient to exclude those tissues. It was also determined that pixels at the edge of the brain introduced large variations in perfusion values—presumably due to partial volume and susceptibility effects, and thus were also excluded. Perfusion values were determined without the knowledge of which groups the rats belonged to avoid bias.

**Neurological function assessment**. The rat's forelimb deficits were evaluated by forelimb-to-wall contact time in the cylinder test. The rat's balancing ability was determined by recording the angle at which the animal started to slip on a rising

inclined plane. Abnormalities in the rat's hind limbs were assessed by scoring the severity of hind limb clasping. For the rotarod test, animals were acclimatized to the static rod 2 days before testing. On the testing day, the speed of the rotarod was increased from 0 to 40 rpm within 2 min. Each rat was tested on the Rotarod for a total of 4 trials per day over 5 consecutive days. For each training day, the smallest value of latency-to-fall for each rat was discarded. The remaining read-outs were averaged, and a group average was calculated for HIP and WT groups. The composite z-score method was used for analyses[21]. For each behavior test, mean and standard deviation were calculated from individual variables collected from longitudinal assessments from animals across the experimental groups. Z-score for each animal in each behavior test was calculated using the following equation:

$$z - \text{score} = \frac{\text{individual variable} - \text{variable mean}}{\text{variable standard deviation}}$$

The composite score for each animal was calculated by averaging z-scores from each test.

**Immunohistochemistry and immunofluorescence**. For immunohistochemistry, formalin-fixed, paraffin-embedded brain tissues from humans and rats were processed using published protocols[15,18,21,25,26,34]. In brief, tissues sections were deparaffinized in xylene 3 times for 5 min. Sections were rehydrated two times in serial dilutions (100%, 95 and 70%) of alcohol and washed one time in PBS. Sections were incubated in 3% $H_2O_2$ (diluted in methanol) for 30 min at RT to block internal endogenous peroxidase. Sections were then incubated in 1× retrieval buffer (S1699; Dako) for 30 min at 95 ℃ in steamer for antigen retrieval. Sections were cooled down to RT and washed one time in 1× PBS. Sections were blocked for 1 h in solution of 15% horse serum, and washes one time in PBS. Sections were incubated with amylin antibody for 24 h. On second day all sections were washed 2 time in 1× TBS. All sections were incubated with biotinylated IgG secondary antibody for 1 h. All sections were washed again and incubated with ABC complex for 30 min. All sections were washed and developed in AEC chromogen. Sections were washed and blocked in 10% NGS for 1 h. Sections were incubated with second primary antibody overnight. On third day, all sections were washed and incubated with AP-conjugated secondary antibody for 1 h. All sections were washed and developed in Stay-green chromogen. All sections were washed and counter stained with hematoxylin and mounted in water-based mounting media. Antibodies against amylin (1:200; T-4157, Bachem-Peninsula Laboratories), GFAP (1:400; 3670S, Cell Signaling Technology), CD68 (1:200; MCA341GA, Biorad,), CD11b (1:200; MCA275GA, Biorad,) and Aβ (1:400; clone 6E10, 803002, Biolegend) were the primary antibodies. Biotinylated IMPRESS horse anti-mouse-AP conjugated IgG (1:100, A3562, Sigma) and biotinylated anti-rabbit IgG (1:300, BA-1100, Vector) were the secondary antibodies. The specificity of the amylin antibody in both human and rat brain tissues was established in previous studies[15,18,21,25,26,34]. Pancreatic tissue from AKO rats was the negative control for amylin.

In immunofluorescence experiments, we used formalin-fixed, paraffin-embedded human brain tissue processed using published protocols[18,21]. In brief, tissue sections were deparaffinized in xylene (three times, 10 min each), rehydrated in serial dilution of alcohol (100%, 95%, 85%), and washed in 1× PBS once. Antigen retrieval was done in citrate buffer (1×, S1699; Dako) (heated for 30 min at 90 ℃). Sections were blocked in blocking buffer (5% NGS, 2% BSA and 0.25% Triton-X in 1×TBS) for 1 h at RT. After blocking, sections were incubated in primary antibodies mixture for 24 h at 4 ℃. Sections were washed in 1× tris-NaCl solution (three times, 5 min each). Sections were incubated in secondary antibodies mixture for one hours at RT. Sections were washed in tri-NaCl as before and incubated in 0.2% Sudan black for 5 min to block auto-immunofluorescence. Section were washed in 1×PBS three times and mounted in mounting media. Sections were imaged after 24 h of mounting. Anti-amylin (1:200; clone E5; SC-377530; Santa Cruz, and 1:200; and T-4157, Bachem-Peninsula Laboratories), Aβ (1:400; clone 6E10, 803002, Biolegend), IL-1β (1:400; ab9722, Abcam), anti-collagen IV (1:500; ab6586; abcam), anti-alpha smooth muscle actin-Alexa Fluor 405 (1:200; ab210128, abcam), anti-caveolin-1 (1:100; sc-894; Santa Cruz, TX), anti-LRP1 (1:500; sc-57351; Santa Cruz), anti-4HNE (1:200; ab46545; abcam) were the primary antibodies. Secondary antibodies were: Alexa Fluor 488 anti-rabbit IgG (A11034; Thermo Fisher), Alexa Fluor 488 conjugated anti-mouse IgG (1:300; A11029; Invitrogen), Alexa Fluor 568 conjugated anti-rabbit IgG (1:200; A11036; Invitrogen), and Alexa Fluor 568 conjugated anti-mouse IgG (1:300; A11004; Invitrogen). Nuclei were counterstained with DAPI mounting media. For triple staining of human brain tissues, smooth muscle actin-Alexa Fluor 405 antibody was added after staining for human amylin and collagen IV and DAPI free mounting media was used. For Thioflavin S staining, after secondary antibody incubation, brain slides were incubated in 0.5% Thioflavin S for 30 min at room temperature. Slides were then incubated for 3 min in 70% ethanol, 5 min in 0.2% Sudan black before washing and mounting.

**Image analysis**. For IHC analyses, the immunoreactivity intensity signals for each antibody (amylin, Aβ) were deconvoluted and analyzed by using the Color Deconvolution plugin in ImageJ. Vector determination (RGB profile) and threshold were established for each color of interest using images of the single stain (i.e., the positive control for each tested antibody). The threshold in each image was set to visually minimize the background. The established RGB profile and threshold

were applied to a macro script command. The macroscript command was applied to the experimental groups and the percentage of area positive for the color of interest was measured. For estimated cerebrovascular amylin-Aβ deposition, clearly-defined blood vessels that were positive for amylin, Aβ or both were counted. Number of blood vessel counts were normalized to the total imaging area.

**High resolution stochastic optical reconstruction microscopy (STORM)**. Freshly isolated brain capillaries were stained for the STORM using primary antibodies (anti-human amylin; T-4157; Peninsula laboratories and anti-glycophorin A; sc-71159 Santa Cruz biotech) and secondary antibodies (rabbit IgG-atto 647 and mouse IgG-atto 488) as follows: Glass bottom 35 mm dishes (P35G-0.170-14-C, MatTek Corporation) were cleaned with 1 M KOH for 15 min on sonicator, rinsed with Milli-Q water and sterilized at least 30 min under UV light. Freshly isolated brain capillaries were plated on the above dishes for 30 min at ambient temperature and washed with 500 μl of 1× PBS. Capillaries were fixed with 200 μl of 3%PFA + 0.1% glutaraldehyde for 10 min at ambient temperature and reduced with 200 μl of 0.1% NaBH4 for 7 min at ambient temperature. Capillaries were washed with PBS three times and blocked with 200 μl of blocking buffer (3% BSA + 0.2% Triton X-100 in PBS) for 120 min at ambient temperature on the rocker. Then 200 μl of primary antibodies cocktail was added for 60 min at ambient temperature on the rocker and washed five times with 200 μl of washing buffer (0.2% BSA + 0.05% Triton X-100 in PBS) for 15 min per wash at ambient temperature on the rocker. After washing 150 μl of secondary antibodies cocktail was added for 30 min at ambient temperature and washed three times with 200 μl of washing buffer for 10 min each wash on the rocker. Capillaries were postfixed with 200 μl of 3%PFA + 0.1% glutaraldehyde for 10 min at ambient temperature and stored in 500 μl of PBS and kept in STORM imaging buffer [7 μl GLOX (oxygen scavenger) and 70 μl of MEA (Cysteamine hydrochloride) + 620 μl of buffer B (Tris-HCL/NaCl)] during imaging using a Nikon N-SIM N-STORM (Nikon) microscope.

**Proximity ligation assay**. Formalin-fixed and paraffin-embedded sections (10 μm) from brain were rehydrated and pretreated with 95% formic acid for 3 min at ambient temperature to expose antigens. After rinsing in 50 mmol/L Tris-HCl buffer with 150 mmol/L NaCl (pH 7.4), brain sections were incubated with primary antibodies anti-human amylin (anti-human amylin; T-4157; Peninsula laboratories), and mouse anti-Aβ antibody 6E10 (803002, Biolegend) overnight at 4 ℃. Duolink in situ PLA (Duolink In situ PLA, DUO92004, Sigma, USA) was performed according to the published protocol[26]. Briefly, for detection of primary antibody pairs, sections were incubated for 90 min with oligonucleotide-conjugated anti-mouse IgG MINUS and anti-rabbit IgG PLUS (PLA probes) diluted 1:6 in Tris-buffered saline at 37 ℃. Amplified DNA strands were detected with oligonucleotides conjugated to a fluorophore, and nuclei stained with Hoechst dye.

**Flow cytometry**. Each blood sample (100 μl) was incubated with a mixture of CD14 (Abcam, ab203294, 1:400), and human amylin (Santa Cruz, SC-377530, 1:100) primary antibodies for 30 min at ambient temperature. Blood was lysed in 2 ml of lysis buffer (1× BD lysis solution, Cat# 349202, BD Biosciences) for 5 min and was washed in 1×PBS with 4% fetal calf serum (FCS). The remaining cells after washing were incubated in secondary antibodies for 30 min at ambient temperature, then were washed and resuspended in 0.5 ml of 1×PBS with 4% FCS. Secondary antibodies were Alexa Fluor 488 goat anti Mouse IgG (H + L) (cat# A11029; Invitrogen, dilution 1:200 in 1×PBS with 4% FBS) for human amylin, and Alexa Fluor 568 goat anti Mouse (cat# A11004, Invitrogen, dilution 1:200, in 1xPBS with 4% FBS), for CD14, respectively. Flow analysis was done on Cytometers BD Symphony A3. The remaining cells were gated on SSC-A vs. FSC-A to determine live cell populations. Doublets (aggregated cells) were gated out using FSC-H vs. FSC-A. Non-aggregated cells were further gated to select monocytes that were positive for CD14 and/or human amylin. Negative control (no antibody) and positive control (human monocytes) were used to set the upper and lower boundaries (Supplemental Fig. S4).

Data was acquired using BDFacsDiva and analyzed using FlowJo v10 software.

**Tissue homogenization**. Frozen human brain tissues were homogenized in homogenate buffer (150 mM NaCl, 50 mM Tris-HCl, 50 mM NaF, 2% Triton X-100, 0.1% SDS, 1% (v/v) protease and phosphatase inhibitors, pH 7.5). Homogenates were centrifuged at 17,000 × g for 30 min at 4 ℃. The supernatant was separated from pellet after centrifugation and was then used for all experiments. Frozen rat brain samples were homogenized with 1% Triton buffer (25 times tissue volume) containing 20 mM Tris-HCl, 150 mM NaCl, 1 mM EDTA, 1 mM EGTA, 1% Triton X-100 (v/v), 1% (v/v) protease and phosphatase inhibitors, pH 7.5. The homogenates were left on ice for 15 min. The homogenates were centrifuged at 22,000 × g for 15 min at 4 ℃. The supernatant (Triton-soluble fraction) was separated from the pellet and used for all experiments.

**ELISA**. Amylin ELISA (EZHA-52K, Millipore Sigma) was used to measure amylin concentrations in whole blood lysates, plasma, capillary lysates and brain homogenates. Insulin concentration was measured in whole blood lysates by using a sandwich ELISA Kit (AYQ-E10465, Assay Solution). Aβ42 concentrations in brain

homogenates were measured using the sandwich ELISA Kit for $A\beta_{42}$ (Thermo Fisher Scientific, cat # KHB3441). ELISAs for IL-1β (ab255730, abcam), arginase 1 (MBS289817, MyBioSource) and arginase 2 (MBS7216305, MyBioSource) were used according to the manufacturer's protocols. Concentrations of target proteins were normalized to the amount of total protein input assessed using the BCA method.

**Arginase activity assay**. Arginase activity was measured in brain microvessel lysates using a colorimetric assay (MAK112, Sigma, MO, USA). The assay was performed according to the published protocol[34]. Briefly, samples were incubated with 10 μl of 5× substrate buffer for 2 h at 37 °C. No substrate was added to sample blank wells. To stop the arginase reaction 200 μl of the Urea reagent was added to each well (Urea standard, water, samples and sample blank). Then 10 μl of 5× substrate buffer was added to sample blank wells and incubated at 37 °C for 60 min. Absorbance reading was taken at 430 nm. Arginase activity was calculated based on the manufacturer's analysis instructions.

**Nitrite and nitrate assays**. Nitrite assay (M36051, Molecular probes) and Nitrate assay (ab65328, Abcam) of rat plasma samples were performed according to the manufacturer's protocol. Briefly, for Nitrite assay, the first 100 μl of working nitrite quantitation reagent was loaded on a microplate, and then 10 μl of nitrite standard and plasma samples were added in duplicates. The microplate was incubated for 10 min at ambient temperature and then 5 μl of nitrite quantitation developer was added to each well. After optimum color development fluorescence intensities were measured at 365/450 nm (excitation/emission). For Nitrate assay, the first 85 μl of nitrate standards and 85ul of plasma samples were loaded on microplate in duplicates, then 5 μl of nitrate reductase and 5 μl of enzyme cofactor were added to each standard and sample well. The plate was incubated for 1 h at ambient temperature. Then 5 μl of enhancer was added to each standard and samples well followed by 50 μl of Griess reagent R1 and R2. Output intensities were measured on a microplate reader at 540 nm.

**Brain capillaries isolation**. Brain capillaries were isolated from rat brains using the published protocol[18]. Brains were homogenized in ice cold PBS and were mixed with 30% Ficoll solution by gently inverting tubes and then centrifuged at $5800 \times g$ for 20 min at 4 °C. Pellets were re-suspended in ice cold PBS with 1% BSA and were passed through the glass beads column. Beads were agitated in PBS with 1% PBS using a 25 mL pipette and solutions were divided equally in two 50 mL tubes. Tubes were centrifuged and pellets were dissolved in 0.5 to 1 mL PBS and used for experiments.

**Smooth muscle cells isolation**. Midcerebral arteries were removed, cleaned of connective tissue, and placed in digestion buffer containing (in mmol/L) 130 NaCl, 1 KCl, 0.2 CaCl₂, 0.5 MgCl₂, 0.33 NaH₂PO₄, 3 pyruvate, 25 HEPES, and 22 glucose (adjusted to pH 7.4 with NaOH). Arteries were then incubated for 15 min at 37 °C in digestion buffer supplemented with papain (0.5 mg/mL) and dithiothreitol (1 mg/mL) followed by a second incubation (15 min at 37 °C) in digestion buffer supplemented with Sigma collagenases Type F and Type H (1 mg/mL each). Arteries were then washed with digestion buffer and kept on ice for 15 min after which gentle agitation with a fire-polished Pasteur pipette was used to create a cell suspension.

**Arterial tone of isolated cortical arteries**. Arterial tone was measured in freshly isolated pial and posterior cerebral arteries using an IonOptix Vessel Diameter system and published protocol[36]. Briefly, arteries were isolated from a fresh rat brain and dissected free from connective tissues. Branches were tied off in HEPES-PSS (141.9 mM NaCl, 4.7 mM KCL, 1.7 mM MgSO4, 0.5 mM EDTA, 2.8 mM CaCl2, 10 mM HEPES, 1.2 mM KH2PO4and 5 mM glucose). One end of artery was cannulated over micropipette within the experimental chamber of the myograph and the other end was blind-closed. Absence of leakage was verified by the maintenance of a stable pressure. Initially intraluminal pressure was kept at 10 mmHg for 15 min, then intraluminal pressure was increased from 20 to 60 mmHg (20 mmHg→40 mmHg→ 60 mmHg→80 mmHg-→100 mmHg-→120 mmHg) stimulating the development of arterial tone. Stable diameter after each increase in intravascular pressure was recorded as active diameter. Next, HEPES-PSS was replaced with a 0 calcium and 2 mM EGTA to determine the arterial passive diameter. Intraluminal pressure was reduced in the opposite direction and maximum passive diameter was recorded for each pressure point. Arterial tone was calculated using the following formula: Arterial tone = ((Passive diameter-active diameter)/passive diameter) *100).

**Immunoprecipitation**. To immunoprecipitate Aβ from brain homogenates and plasma, a published protocol was used[21]. Briefly, 1 mg of protein was incubated with anti-rat Aβ (2 μg; CST2454; Cell Signaling Technology) overnight with end-over-end rotation, at 4 °C. Antigen-antibody complex was added to Immobilized Protein A/G resin slurry for 2 h at ambient temperature, washed and samples eluted from the resins using elution buffer. The eluate was used for Western blot analysis. Amersham ECL Rabbit IgG, HRP-linked whole Ab (from donkey)

(Cytive, Cat# NA934V-HRP), against both light and heavy chain IgG, was the secondary antibody (1:20,000 dilution).

**Western blot**. Western blot analysis was performed on isolated brain capillaries, brain tissue homogenate and plasma from rats using published protocols[15,18,21,25,26,34]. RIPA buffer with 2% SDS was used to retrieve Aβ monomers from frozen brain samples. The lysate was centrifuged at $17,000 \times g$ for 30 min. The supernatant was separated from the pellet after centrifugation and then used for Western blotting. Total protein levels were estimated using a BCA kit (23225, ThermoFisher). Anti-LRP1 antibody recognizing the β-subunit of LRP1 (1:1,000; clone 5A6; sc-57351; Santa Cruz), anti- P-glycoprotein or Pgp (1:1000, ab170904, Abcam), rabbit anti-amylin polyclonal (1:2000; T-4157, Bachem-Peninsula Laboratories, CA), anti-rat and human Aβ (1:1,000; 2454; Cell Signaling Technology), mouse anti-β actin (1:10,000; clone BA3R; MA5-15739; Thermo-Fisher), mouse monoclonal anti-GAPDH (1:10,000; clone 6C5; ab8245; Abcam), anti-rabbit IgG HRP conjugated (1:30,000; NA934VS; GE Healthcare) and anti-mouse IgG HRP conjugated (1:20,000; NXA931; GE Healthcare) were primary antibodies. Immunoprecipitated rat Aβ from homogenates and matched plasma (50 μg of protein from tissue homogenate or immunoprecipitated rat Aβ elution) were loaded on 8% SDS-PAGE gels. Aggregated Aβ from brain homogenates was resolved in native-PAGE (non-reducing; non-denatured). Monomeric Aβ peptides were resolved in acidic Bis-Tris gels with 8 M urea. To enhance the signal for monomeric Aβ, membranes were boiled for 3 min in PBS before the blocking step. LRP1 in cell and brain capillary lysates was resolved using 4–12% Bis-Tris gel under non-reducing conditions. HRP-conjugated anti-rabbit or anti-mouse were secondary antibodies. Equal loading in Western blot experiments was verified by re-probing with a monoclonal anti-β actin antibody (raised in mouse, clone BA3R, Thermo Scientific; 1:2000). Protein levels were compared by densitometric analysis using ImageJ software. Negative control experiments were conducted to test the specificity of the bands identified in Western blotting after immunoprecipitation (Fig. 5b). Briefly, duplicate brain homogenate and plasma samples (1 mg of total protein) were used for immunoprecipitation with: 1, anti-Aβ antibody (CST2454; Cell Signaling Technology); and 2, anti-IgG antibody (Cytive, Cat# NA934V-HRP). After blotting and blocking, both membranes were incubated with the anti-Aβ antibody and further analyzed and imaged together. The results show no or minor immunoreactivity signal intensities in the IgG-immunoprecipitated sample set (Supplemental Fig. S5) demonstrating the specificity of the Aβ antibody.

**Amylin aggregation and injection**. Lyophilized amidated human amylin peptide (Anaspec #AS-60254-1) was dissolved in PBS pH 7.4 to the concentration of 50 μM. The mixture was incubated at 37 °C for 72 h with occasional shaking to allow amylin to form aggregates. Aggregated human amylin solution was injected into 9–10 months old AKO rats via tail vein (60 μg/kg) ($n = 3$ rat males), daily via tail vein for 1 week.

**In vitro BBB model of $A\beta_{42}$ transcytosis with amylin-induced stress at the blood side**. We used an in vitro BBB model with amylin deposition on the EC monolayer and consequent effects on the Aβ transport across the EC monolayer. Briefly, primary rat brain microvascular endothelial cells (Cell Applications Inc) were plated on 24-well Transwell-Clear inserts with 0.4 μm pore polycarbonate membrane (Costar, Corning, NY, USA) and primary rat brain astrocytes (Sigma) were cultured at the bottom wells. Barrier integrity was measured from the trans-endothelial electrical resistance (TEER) using the EVOM2 meter with STX-3 electrodes (World Precision Instruments). Maximum TEER was achieved within 8–10 days in culture. BBB permeabilities to human amylin (10 μM; Anaspec; AS-60254-1), rat amylin (10 μM; American Peptide) and DMSO (1 mM; vehicle) were assessed using FITC-Dextran 4 kDa (Fisher Scientific) diluted in Hank's Balanced Salt Solution (HBSS) buffer with 0.1% BSA (HBSS-BSA) as a paracellular diffusion marker. Permeability coefficients were calculated using the formula; $P = (\Delta Q/\Delta t)/(A*C_0)$, $(\Delta Q/\Delta t) =$ rate of FITC-Dextran change; $A =$ surface area of insert (0.33 cm²); $C_0 =$ Initial FITC-Dextran input.

In the $A\beta_{42}$ transcytosis experiments, the EC monolayer was incubated with human amylin (10 μM) or vehicle (DMSO) for 24 h. After washing, the luminal chamber was replaced with HBSS-BSA, and the abluminal chamber with $A\beta_{(1-42)}$-FAM (5 μM; Bachem) or FITC-Dextran, respectively. $A\beta_{(1-42)}$ samples were collected from the luminal chamber for the measurement of the $A\beta_{(1-42)}$ − FAM and FITC-Dextran fluorescence intensities and $A\beta_{(1-42)}$ transcytosis quotient (TQ) as described previously [38]: $TQ = (A\beta_{(1-42)} − FAM_{luminal}/A\beta_{(1-42)} − FAM_{input})/(FITC-Dextran_{luminal} − FITC-Dextran_{input})$.

**MTS cytotoxicity assay**. CellTiter 96® AQueous One Solution Cell Proliferation Assay (MTS) (Promega) was used to assess cytotoxicity of amyloid-forming human amylin on the EC monolayer.

**Real-Time Quantitative Reverse Transcription PCR**. Total RNA was isolated using RNAqueous total RNA isolation kit according to the manufacturer's protocol (Invitrogen, AM1914). cDNA synthesis and amplification were done using iTaq Universal SYBR Green One-Step Kit (Biorad; 1725151) with the following primer

sequences: *LRP1*: forward (Fwd) 5′-TTGTGCTGAGCCAAGACATC-3′, reverse (Rev) 5′-GGCGTGGAAGACATGTAGGT-3′; and *GAPDH*: Fwd 5′- GCTGCG TTTTACACCCTTTC-3′, Rev 5′-GTTTGCTCCAACCAACTGC-3′ (IDT, Inc, USA). For miRNA quantification, cDNA was synthesized from total RNA using miRNA cDNA synthesis kit with poly (A) polymerase (ABMgood, G902). cDNA was amplified using SYBR Green mastermix (Biorad) along with miRNA specific primers from (*rno-miR-103-3p*, MPR00332; *rno-miR-107-3p*, MPR00335; *RNU6* house Keeping gene, MP-r99998) (ABMgood). Data were analyzed using the $2^{-\Delta\Delta Ct}$ method, and experiments were normalized to *GAPDH* or *U6 miRNA*.

**MicroRNA mimics and antagomir transfection.** To study the role of miRNA signaling in amylin-induced suppression of endothelial *LRP1* expression, we used transfection of rat brain microvascular ECs with *miR-103–3p* (MCR01039), *miR-107-3p* (MCR01045) and control (MCH00000) mimics (ABMgood). Antagomir *miR-103-3p* (IH-320345-05-0005), *miR-107-3p* (IH-320348-05-0005) and negative control (IN-001005-01-05) (Dharmacon Inc.) were used in an attempt to rescue LRP1 expression. All transfections were done using RNAiMAX (Invitrogen) as per the manufacturer's recommended protocol. Briefly, ECs were plated at 50% confluency in six-well plates followed by co-transfection with either 100 nM of *103-3p* and *107-3p* mimics or antagomirs along with their respective negative controls. After 12 h, antagomir-treated cell groups were further treated with 10 μM human amylin for 24 h. After 36 h of transfection, cells were harvested for Western blot analysis.

**Human amylin aggregation curve and the Thioflavin T assay for amyloid.** Lyophilized human amylin (above) was reconstituted in 20 mM Tris-HCl (pH 7.4) to 100 μM. Different concentrations were made with Tris-HCl. At each time point, 50 μL of each was added in 96 well plate with 50 μL 0.016 mg/mL (50.17 mM) Thioflavin T (Sigma). Red blood cells (RBCs) were diluted 1:10 with PBS. In each well, 50 μL of diluted RBC/capillary was added. Then 50 μL of 0.016 mg/mL ThioT solution (made in PBS) was added to each well. Fluorescence at 437 nm excitation and 485 nm emission was measured. The final concentration of ThioT optimized for signal intensity detection was 50 μM. The results were normalized to the total protein input (RFU/μg- Relative Fluorescence Unit/μg). Data are shown as fold changes compared to WT (or APP/PS1) rats because the experiment is not a quantitative test (no peptide standards).

**Lipid peroxidation and reactive oxygen species (ROS).** Lipid peroxidation was measured in isolated smooth muscle cells and cultured rat brain microvascular ECs using the published protocol[51]. In brief, cells were incubated with the fluorescent probe 4,4-difluoro-5-(4-phenyl-1,3-butadienyl)-4-bora-3a,4a-diaza-s-indacene-3-undecanoic acid (C11-BODIPY581/591; D3861; Invitrogen; OR) and Liperfluo for 20 min at ambient temperature. After staining cells were washed four times and analyzed with a fluorescence microscope.

**RNAseq analysis.** Gene-level differential expression between HIP and WT rats ($n = 10$ males/group) was analyzed to test whether brain genes respond directly to amylin-mediated cerebrovascular inflammation following the development of cerebrovascular amylin deposits. Cerebral cortex RNA was isolated using Qiagen's RNeasy Mini Kit (Ref # 74104). Omega Bioservices performed RNAseq library preparations and sequencing using the Illumina HiSeq 2500. RNAseq data were analyzed using PartekFlow software (Partek, MO). Briefly, RNAseq fastq files were imported, aligned to rat reference genome (Rattus norvegicus-rn7) and quantified at gene level using Ensembl105 annotation. RNAseq read counts were normalized and further analyzed for gene differential expression between HIP and WT groups using DESeq2 algorism. RNA transcripts abundance of 403 genes had $P$-values ≤ 0.05 between the HIP and WT groups. These differentially expressed (DE) genes were queried for an enrichment of canonical pathways using Ingenuity Pathway Analyses software (IPA, Qiagen) and further for enrichment in Gene Ontology (GO) biological processes of these DE genes using DAVID (NIH).

**Statistics and reproducibility.** The number of samples or animals in each analysis, the statistical analysis performed and $P$ values are reported in figures and figure legends. Sample size for animal behavior study was determined based on pilot study. Sample size was calculated using Excel with known standard deviation, confidential interval of 95%, and alpha equals 0.05. D'Agostino–Pearson and Kolmogorov–Smirnov test was used to test normality distribution of continuous variables. Parametric comparisons of continuous variables with normal distributions were performed using two-tailed unpaired t-test. Welch's correction was used with t-test to account for unequal variance from unequal sample sizes, if necessary. Parametric comparisons of three groups or more group means were performed using one-way or two-way ANOVA with the Bonferroni post-test. Kruskal–Wallis one-way analysis of variance was used for analyses of continuous variables with skewed distributions and similar spread across the groups (as in Supplemental Fig. S1a). Relationships between two continuous variables were analyzed by correlation analysis. Data are presented as mean ± S.E.M or as box and whisker plots. Difference between groups was considered significant when $P < 0.05$. All analyses were performed using GraphPad Prism 8.1.

Replication was performed at different time point to verify the reproducibility of date obtained. ELISAs and Western blots were conducted in duplicates by two different team members. The data were reproducible upon replication. Observer-blind analyses were performed on all human tissue samples; the results were communicated to the AD Center at University of Kentucky to assess the relationships with AD pathology/cognitive function. Investigators were also blinded in the scoring of histological analyses and in the longitudinal animal behavior testing to prevent bias. Investigators were not blinded during pharmacological intervention (such as amiR treatments of EC cells) or injection (human amylin into AKO rats) because the interventions were performed by the same personnel. Investigators were not blinded during most of biochemical assays because it is not necessary. Data collection was performed at the same time for experimental groups with the same setting.

**Validation of antibodies.** All of antibodies used in this manuscript are well-established by manufacturers and other publications. We performed validation for amylin antibody in this study.

**Reporting summary.** Further information on research design is available in the Nature Portfolio Reporting Summary linked to this article.

## Data availability
Data supporting the findings of this study are available in within the article and its supplementary information files. Source data underlying graphs are available as 'Supplementary data figure' files. Uncropped and unedited bot images are available as Supplemental Fig. 6. Source data for all the figures are provided in a single excel file as 'Supplementary Data 1'. RNAseq data discussed in this publication have been deposited in the NCBI's Gene Expression Omnibus and are accessible through GEO Series accession number GSE221450.

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

## Acknowledgements

Funding in part by: University of Kentucky Research Alliance to Reduce Diabetes-Associated Microvascular Dysfunction (ADAM) and National Institutes of Health R01 NS116058, R01 AG057290, R01 AG053999, R01 HL 149127 and P30 AG028383, and Alzheimer's Association VMF-15-363458. UK Dementia Research Institute which receives its funding from DRI Ltd, funded by: the UK Medical Research Council, Alzheimer's Society and Alzheimer's Research UK; Medical Research Council (award number MR/N026004/1); Wellcome Trust Hardy (award number 202903/Z/16/Z); Dolby Family Fund; National Institute for Health Research University College London Hospitals Biomedical Research Center; BRCNIHR Biomedical Research Center at University College London Hospitals NHS Foundation Trust and University College London. H.L. was supported by an American Heart Association fellowship (18PRE33990154). T.L. is supported by an Alzheimer's Research UK Senior Fellowship. H.Z. is a Wallenberg Scholar supported by grants from the Swedish Research Council (#2018-02532), the European Research Council (#681712 and #101053962), Swedish State Support for Clinical Research (#ALFGBG-71320), the Alzheimer Drug Discovery Foundation (ADDF), USA (#201809-2016862), the AD Strategic Fund and the Alzheimer's Association (#ADSF-21-831376-C, #ADSF-21-831381-C and #ADSF-21-831377-C), the Olav Thon Foundation, the Erling-Persson Family Foundation, Stiftelsen för Gamla Tjänarinnor, Hjärnfonden, Sweden (#FO2019-0228), the European Union's Horizon 2020 research and innovation program under the Marie Skłodowska-Curie grant agreement No 860197 (MIRIADE), the European Union Joint Program – Neurodegenerative Disease Research (JPND2021-00694), and the UK Dementia Research Institute at UCL (UKDRI-1003).

## Author contributions

N.V. – immunohistochemistry, flow cytometry, confocal microscopy, STORM, brain capillary and artery isolation; G.V.V. – LRP1 Western blot analysis and in vitro BBB experiments; E.W. – flow cytometry and brain tissue processing for RNAseq analyses; D.K., N.L., and L.R. – rat genotyping, tissue collection and processing, ELISA; H.C. – rat behavior, Th-T assay in rat blood; D.K.P. and J.H.W. – MRI and ASL experiments; K.C.C. - RNAseq data analysis; S.D. – SMCs isolation and confocal microscopy analysis of oxidative stress in isolated SMCs; A.M.S. – flow cytometry training of E.W. and flow cytometry data analysis; L.V.E. – neuroinflammation experimental protocol; A.J.M. – vascular arginase-NO regulation experimental protocol; M.F.N. and M.A.N. - pressure myography experiments; G.A.J., P.T.N., D.M.W., L.V.E., H.Z., C.T., T.L., and J.H. – provided human samples; G.A.J., L.G.B, P.T.N., J.H., H.Z., T.L., G.J.B., and F.D. – interpretation of amylin-Aβ pathology; F.D. – conceptualization, resources, data analysis and drafted the manuscript, all other authors contributed to the final form of the manuscript.

## Competing interests

The authors declare no competing interests.
