## [Peer Review File · Communications Biology]

Reviewers' comments:

Reviewer #1 (Remarks to the Author):

This MS addresses the important question of whether the pancreatic hormone amylin can contribute to the amyloid burden of brain usually attributed to beta amyloid peptide. The Introduction does a good job of explaining why this is relevant, the previous work relating to this topic (BBB transport of the two peptides, their sources, previous work showing deposition, etc), and the hypothesis being tested here. Important findings include: blood amylin levels are higher in demented persons (see* below); amylin associates with circulating macrophages; correlations of brain amylin levels with blood amylin and brain beta amyloid levels; a differential distribution at the BBB of the two peptides; confirmation of these findings in rats that overexpress human amylin (HIP rats); HIP rats also had cerebrovascular tonic-response abnormalities and amylin-beta amyloid deposits in brain; effects of amylin on beta amyloid transport.

1) *the authors show that glucose levels did not differ between their demented and non-demented groups. But considering that amylin is co-secreted with insulin, it would be interesting to know what insulin levels were in these groups. It's always difficult to know when to measure insulin, but in this study, it should be in the same blood sample used to measure amylin.

2) In the HIP rat, is the amylin exclusively from pancreas or do other sites excrete as well? Is there production of human amylin in brain of the HIP rat?

3) Two issues re: the "Disrupted A β transport across the BBB..." section. First, APP/PS1 may not be a good control as every animal model of AD in which it has been examined has a decreased efflux rate of beta amyloid peptide. Second, the decreased plasma-to-brain ratio for beta amyloid can have causes other than decreased transport, such as altered degradation rates of beta amyloid, increased aggregation causing immunological hiding of beta amyloid, enhanced influx of beta amyloid, or increased blood levels of beta amyloid. Because the authors did not directly measure efflux of beta amyloid and did not rule out other causes of an altered ratio, this is currently an overstatement. Although western blotting supports their conclusion, the literature contains mismatches between protein levels and activity levels for P-gp and LRP-1.

4) Re: effects of amylin on beta amyloid efflux, it would be interesting to determine whether amylin is a competitive inhibitor of beta amyloid efflux as mediated by P-gp or LRP-1.

Reviewer #2 (Remarks to the Author):

Verma and colleagues tested the hypothesis that amylin secreted from the pancreas disturbs brain A β clearance. They used human samples as well as transgenic rat models and defined how pancreatic amyloid-forming human amylin affects brain A β clearance. They found that blood amylin concentrations positively correlated with cognitive impairment. Their data suggest that circulating amylin induced systemic inflammation, cerebrovascular amylin deposits and local perivascular inflammation. They postulated the mechanisms involved a significant reduction of LRP1-mediated A β transport across BBB and clearance through interstitial fluid drainage. Based on these findings, they conclude that altering pancreas-derived amylin in blood significantly influences cerebrovascular amylin deposits and A β pathology.

Overall, the manuscript is well written, and the data presented are high quality.

My major concern is that the authors drew their conclusions largely based on correlative results. For example, the protein levels of both p-gp (Fig 4I) and LRP1 (Fig 4J) were significantly reduced in HIP samples as compared to WT controls. To establish that either p-gp or LRP1 alone, or both, plays a key role in the mediating the effect of amylin on A β clearance, it seems to me, at minimum, the authors

would want to carry out some types of rescuing experiments. I fully understand, to demonstrated this in vivo is beyond of the scope of the current study. Some kinds of in vitro validation will be sufficient here.

The other issue that needs clarification involves the results of Fig 4C, I understand the authors IPed A β from either plasma or brain tissues of WT and HIP samples. One thing I am puzzled is that the major band in both cases, when blotted for A β , is the diffused species of \sim 75-120 kD species. How confident the authors are that this is really "A β "? especially, given that the authors showed that the A β species in APP/PS1 samples in Fig 4B (bottom) were predominantly with much lower molecular weight.

Reviewers' comments:

Reviewer #1 (Remarks to the Author):

This MS addresses the important question of whether the pancreatic hormone amylin can contribute to the amyloid burden of brain usually attributed to beta amyloid peptide. The Introduction does a good job of explaining why this is relevant, the previous work relating to this topic (BBB transport of the two peptides, their sources, previous work showing deposition, etc), and the hypothesis being tested here. Important findings include: blood amylin levels are higher in demented persons (see* below); amylin associates with circulating macrophages; correlations of brain amylin levels with blood amylin and brain beta amyloid levels; a differential distribution at the BBB of the two peptides; confirmation of these findings in rats that overexpress human amylin (HIP rats); HIP rats also had cerebrovascular tonic-response abnormalities and amylin-beta amyloid deposits in brain; effects of amylin on beta amyloid transport.

1) *the authors show that glucose levels did not differ between their demented and non-demented groups. But considering that amylin is co-secreted with insulin, it would be interesting to know what insulin levels were in these groups. It's always difficult to know when to measure insulin, but in this study, it should be in the same blood sample used to measure amylin.

Response: Thank you for the great suggestion. We assessed the amylin-insulin relationship in the same blood samples as in Fig. 1C. The results of insulin and amylin ELISAs show that increased blood insulin concentrations are associated with greater blood amylin concentrations ($r = 0.52$; $P < 0.0001$) (Fig. 1D and Supplemental Fig. S1C). The pairwise correlation coefficient demonstrates that hyperamylinemia and hyperinsulinemia are correlated in AD dementia (lines: 118-123).

2) In the HIP rat, is the amylin exclusively from pancreas or do other sites excrete as well? Is there production of human amylin in brain of the HIP rat?

Response: In our previous report (Ref. 25; PMID: 25149184), the specificity of human amylin RNA expression in the pancreas and lack of human amylin RNA in the brain in HIP rats was documented by qRT-PCR. Our data showed that the mRNA level of human amylin in the HIP rat pancreas is comparable to that in human pancreas and that human amylin mRNA in the HIP rat brain is below the limit of detection. This clarification was added in the revised manuscript (lines: 187-189).

3) Two issues re: the "Disrupted Abeta transport across the BBB..." section. First, APP/PS1 may not be a good control as every animal model of AD in which it has been examined has a decreased efflux rate of beta amyloid peptide.

Response: APP/PS1 rats were **not** controls for efflux rate of A β .

Brain homogenate from a 12-month-old APP/PS1 rat was used as the positive control for: 1, Western blotting of A β in HIP rat brain homogenate (Fig. 5A; revised manuscript); and 2, immunoprecipitation experiments to enrich A β in plasma samples and brain homogenates from age-matched HIP and WT rats (Fig. 5B; revised manuscript). We also used APP/PS1 rats with pancreatic expression of human amylin (APP/PS1/HIP rats) to test the hypothesis that the

secretion of amyloid-forming human amylin from the pancreas into the blood leads to formation of cerebrovascular amylin-A β deposits (Fig. 3D-F; revised manuscript).

Second, the decreased plasma-to-brain ratio for beta amyloid can have causes other than decreased transport, such as altered degradation rates of beta amyloid, increased aggregation causing immunological hiding of beta amyloid, enhanced influx of beta amyloid, or increased blood levels of beta amyloid. Because the authors did not directly measure efflux of beta amyloid and did not rule out other causes of an altered ratio, this is currently an overstatement. Although western blotting supports their conclusion, the literature contains mismatches between protein levels and activity levels for P-gp and LRP-1.

Response: We appreciate very much this comment. Within the revised version of the manuscript, alternative explanations of the plasma-to-brain ratio for A β results are discussed in the final paragraph of the “**Antisense microRNAs rescue of amylin-induced LRP1 suppression**” subsection (lines: 322-337).

4) Re: effects of amylin on beta amyloid efflux, it would be interesting to determine whether amylin is a competitive inhibitor of beta amyloid efflux as mediated by P-gp or LRP-1.

Response: To determine whether amyloid-forming amylin is a competitive inhibitor of A β efflux as mediated by P-gp and LRP1, we employed an *in vitro* model of BBB in which the EC monolayer was exposed to amyloid-forming human amylin on the “luminal side” and A β at the “abluminal” (brain side), as shown in Fig. 6 (revised manuscript). The presence of amyloid-forming human amylin at the “luminal side”, reduced the A β transcytosis quotient by $20 \pm 5\%$ ($P < 0.05$). The results indicate that amyloid-forming amylin directly influences A β efflux through suppressing A β transport protein expression at a post-transcriptional level. Please see the subsection “*In vitro* BBB model of A β transcytosis with amylin-mediated EC endothelial stress”, for detail. Additional mechanistic experiments in cell models were included in a new paragraph “*Antisense microRNAs rescue of amylin-induced LRP1 suppression*” (lines: 322-337).

Reviewer #2 (Remarks to the Author):

Verma and colleagues tested the hypothesis that amylin secreted from the pancreas disturbs brain A β clearance. They used human samples as well as transgenic rat models and defined how pancreatic amyloid-forming human amylin affects brain A β clearance. They found that blood amylin concentrations positively correlated with cognitive impairment. Their data suggest that circulating amylin induced systemic inflammation, cerebrovascular amylin deposits and local perivascular inflammation. They postulated the mechanisms involved a significant reduction of LRP1-mediated A β transport across BBB and clearance through interstitial fluid drainage. Based on these findings, they conclude that altering pancreas-derived amylin in blood significantly influences cerebrovascular amylin deposits and A β pathology.

Overall, the manuscript is well written, and the data presented are high quality.

My major concern is that the authors drew their conclusions largely based on correlative results. For example, the protein levels of both p-gp (Fig 4I) and LRP1 (Fig 4J) were significantly reduced in HIP samples as compared to WT controls. To establish that either p-gp or LRP1 alone, or both, plays a key role in the mediating the effect of amylin on A β clearance, it seems to me, at minimum, the authors would want to carry out some types of rescuing experiments. I fully

understand, to demonstrated this *in vivo* is beyond of the scope of the current study. Some kinds of *in vitro* validation will be sufficient here.

Response: Thank you for the excellent suggestion. We employed an *in vitro* BBB model of A β transcytosis in which the EC monolayer was exposed to amylin-mediated stress; antisense microRNAs were used in an attempt to rescue endothelial LRP1 expression (Fig. 6). The *in vitro* BBB model of A β efflux under amylin-induced EC oxidative stress shows that downregulated LRP1 is a miRNA-based translational repression mechanism that can be partly reversed by antisense microRNA. The experimental approach and results are included in a new paragraph “*Antisense microRNAs rescue of amylin-induced LRP1 suppression*” (lines: 322-337).

The other issue that needs clarification involves the results of Fig 4C, I understand the authors IPed A β from either plasma or brain tissues of WT and HIP samples. One thing I am puzzled is that the major band in both cases, when blotted for A β , is the diffused species of ~75-120 kD species. How confident the authors are that this is really “A β ”? especially, given that the authors showed that the A β species in APP/PS1 samples in Fig 4B (bottom) were predominantly with much lower molecular weight.

Response: Additional negative control experiments were conducted to test the specificity of the bands identified in Western blotting after immunoprecipitation (Supplemental Fig. S5). Briefly, duplicate brain homogenate and plasma samples (1mg of total protein) were used for immunoprecipitation with: 1, anti-A β antibody; and 2, anti-IgG antibody. After blotting and blocking, both membranes were incubated with the anti-A β antibody and further analyzed and imaged together. The results show no or minor immunoreactivity signal intensities in the IgG-immunoprecipitated sample set demonstrating the specificity of the A β antibody.

In the Western blotting after immunoprecipitation experiment (Fig. 5B, revised manuscript), we used brain tissue homogenate from an APP/PS1 rat (age, 12-months) as the positive-control for A β . The molecular weight bands of enriched A β in the APP/PS1 brain tissue homogenate are in the same range (75-120 kDa) as those in the enriched A β in HIP and WT brain homogenates suggesting the presence of A β in all samples. In Fig. 5A (revised manuscript) (Fig. 4B in the initial manuscript), A β species in APP/PS1 samples were predominantly with much lower molecular weight because we used RIPA buffer with 2% SDS to retrieve low molecular weight A β species.

REVIEWERS' COMMENTS:

Reviewer #1 (Remarks to the Author):

Thank you for fully addressing my concerns.

Reviewer #2 (Remarks to the Author):

The authors have addressed all the issues I raised. I have No more concerns.